# UNIVIDEO: UNIFIED UNDERSTANDING, GENERATION, AND EDITING FOR VIDEOS

**Cong Wei**[1]  **Quande Liu**[2†]  **Zixuan Ye**[2]  **Qiulin Wang**[2]  **Xintao Wang**[2]
**Pengfei Wan**[2]  **Kun Gai**[2]  **Wenhu Chen**[1†]

[1] University of Waterloo     [2] Kling Team, Kuaishou Technology

## ABSTRACT

Unified multimodal models have shown promising results in multimodal content understanding and generation but remain largely limited to the image domain. In this work, we present `UniVideo`, a versatile framework that extends unified modeling to the video domain. `UniVideo` adopts a dual-stream design, combining a Multimodal Large Language Model (MLLM) for visual understanding with a Multimodal DiT (MMDiT) for visual generation. This design preserves the MLLM's original text generation capabilities, enables accurate interpretation of complex multimodal instructions, and maintains visual consistency in the generated content. Built on this architecture, `UniVideo` unifies diverse video generation and editing tasks under a single multimodal instruction paradigm and is jointly trained across them. Extensive experiments demonstrate that `UniVideo` competitive or superior state-of-the-art task-specific baselines in visual understanding, image generation and editing, text/image-to-video generation, in-context video generation and in-context video editing. Notably, the unified design of `UniVideo` enables two forms of generalization. First, `UniVideo` supports task composition, such as combining editing with style transfer, by integrating multiple capabilities within a single instruction. Second, even without explicit training on free-form video editing, `UniVideo` transfers its editing capability from large-scale image editing data to this setting, handling unseen instructions, such as changing the environment or altering materials within a video. Beyond these core capabilities, `UniVideo` also supports generation with thinking, where the MLLM interprets complex prompts and guides the MMDiT during synthesis. To foster future research, we released our model and code at https://github.com/KlingTeam/UniVideo.

## 1 INTRODUCTION

A long-term goal of multimodal AI assistants is to build models that can seamlessly **understand** diverse inputs across modalities and **generate** outputs in kind, enabling natural communication through language, images, and video demonstrations.

Recent advances in unified models suggest that this vision is increasingly attainable. Prior work (Shi et al., 2024a; Pan et al., 2025; Sun et al., 2023; Team, 2024; Tong et al., 2024; Wang et al., 2024b; Deng et al., 2025; Wu et al., 2025b; Ma et al., 2025b; Xie et al., 2024; 2025; Zhou et al., 2024) has demonstrated promising results in text–image understanding and generation by jointly optimizing these capabilities within unified systems. More recently, models such as Google Nano banana and GPT-image-1 have pushed this paradigm further by integrating computer vision, image manipulation, and multimodal reasoning into a single framework, marking a shift from specialized single-modality generators toward powerful unified systems.

Despite this progress, unified understanding–generation models remain limited to text and image (Lin et al., 2025; Wu et al., 2025c), leaving video largely underexplored. Existing video generation models primarily address a single text-to-video task and rely on text encoders to process instructions

---

†Corresponding authors.

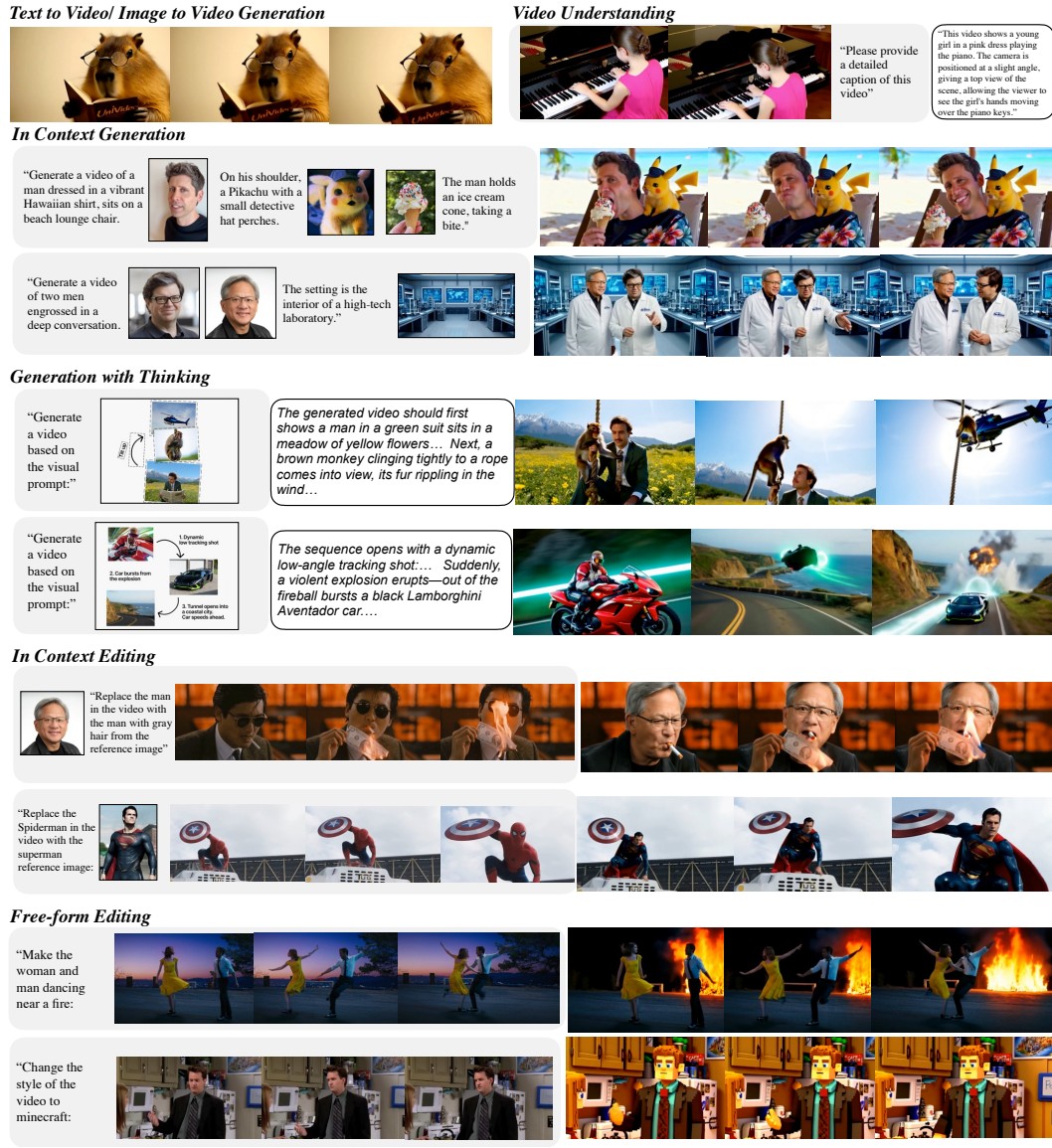

Figure 1: `UniVideo` is a unified system that can **understand** multi-modal instructions and **generate** multi-modal content. More videos are available on https://congwei1230.github.io/UniVideo.

(Wan et al., 2025; Ju et al., 2025b; Polyak et al., 2024; Kong et al., 2024), restricting their ability to understand and reason over multimodal instructions (Hu et al., 2024a). Meanwhile, video editing methods typically employ task-specific modules or pipelines (Ku et al., 2024; Jiang et al., 2025; Ye et al., 2025b), which makes it difficult to scale across diverse tasks. Consequently, due to the lack of unified modeling, advanced capabilities such as multimodal prompting, in-context video generation, and sophisticated free-form editing remain beyond the reach of any single model.

Motivated by these limitations, we present `UniVideo` —a unified framework for understanding, generation, and editing in the video domain. `UniVideo` bridges this gap by enabling multimodal instruction following and delivering robust performance across diverse video tasks.

To build `UniVideo`, we propose a two-stream design, where an MLLM serves as the *understanding branch* and an MMDiT backbone (Esser et al., 2024) serves as the *generation branch*. While prior work such as Qwen-Image (Wu et al., 2025a) explores a similar idea in the image domain, our model generalizes this design to video. Both streams now receive image and video instructions: the understanding branch through a semantic encoder, and the generation branch through VAE-based

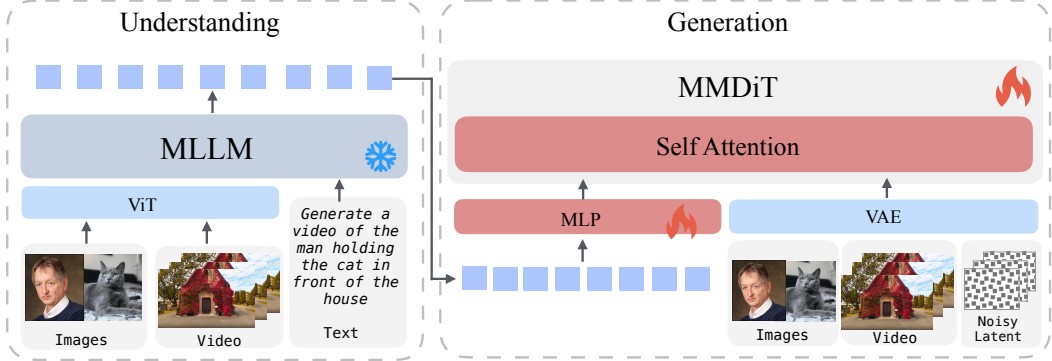

Figure 2: **Model architecture.** `UniVideo` is a dual-stream model consisting of an MLLM for understanding and an MMDiT module for generation. While prior work such as Qwen-Image and OmniGen2, explores a similar idea in the image domain, our model generalizes this design to video.

encoders. In contrast, prior unified models such as GPT-image-1 (Lin et al., 2025) rely exclusively on semantic encoders, which often struggle to capture fine-grained visual details. Similarly, bottlenecked approaches using learnable query tokens (Tong et al., 2024; Pan et al., 2025) compress inputs into a fixed set of tokens, creating a severe capacity bottleneck when instructions contain videos. As a result, both approaches fall short in supporting in-context video generation. Our design preserves the multimodal reasoning capabilities of the MLLM while enabling the model to handle diverse video tasks with multimodal inputs. Moreover, it ensures cross-stream consistency, which is crucial for precise editing and for maintaining subject identity in in-context generation.

Based on this unified architecture, we train `UniVideo` across a wide spectrum of tasks, including text-to-image, text-to-video, image-to-video, in-context video generation, in-context video editing, and image editing. As a unified system, `UniVideo` not only understands multimodal instructions and distinguishes between tasks but also achieves improvements over state-of-the-art task-specific methods. Thanks to unified training, `UniVideo` generalizes to novel task compositions unseen during training, such as deleting one identity while swapping another within a single instruction. More importantly, although `UniVideo` is not trained on free-form video editing data, it demonstrates generalization ability transfer from image editing to free-form video editing (e.g., changing object materials or modifying weather conditions), highlighting the effectiveness of our unified video understanding and generation framework.

Furthermore, `UniVideo` retains the strong visual understanding and text generation capability of its underlying frozen MLLM. By leveraging the MLLM's autoregressive reasoning and language generation abilities, `UniVideo` can effectively interpret ambiguous and complex multimodal instructions that require joint vision–language understanding, such as reasoning over visual prompts before performing generation. Since `UniVideo`'s text generation ability originates from a frozen MLLM, `UniVideo` should be regarded as *a post-trained unified multimodal generative system*(Wu et al., 2025c; Pan et al., 2025) capable of producing images, videos, and text, rather than a unified model trained from scratch(Ma et al., 2025b; Deng et al., 2025).

**Our key contributions are:**
**1)** We introduce `UniVideo`, a multimodal generative model that unifies understanding, generation, and editing of images and videos within a single framework. To build `UniVideo`, we propose a dual-stream architecture that combines the multimodal reasoning capabilities of the MLLM with the generation strengths of the MMDiT. Unlike prior task-specific or modality-restricted approaches, `UniVideo` can interpret multimodal instructions, distinguish between diverse tasks, and achieve state-of-the-art performance across a wide range of benchmarks.
**2)** We systematically study the key design choices that enable this unified framework, including connector architectures, generator designs, and multimodal conditioning strategies, and provide empirical evidence for their effectiveness.
**3)** We demonstrate that `UniVideo` generalizes to unseen tasks and novel task compositions without ad hoc designs, highlighting the benefits of a unified framework.

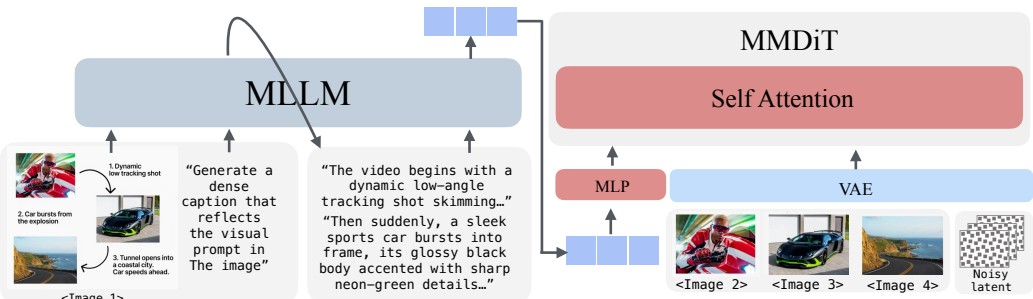

Figure 3: **Generation with thinking.** `UniVideo` leverages the MLLM stream to understand and interpret user intent from complex multimodal prompts that cannot be handled by the DiT alone. For example, users can provide diagrams or visual annotations to guide video generation without writing dense textual prompts.

## 2 METHOD

### 2.1 MODEL ARCHITECTURE

As demonstrated in Figure 2, `UniVideo` consists of two main components: a multimodal large language model (MLLM) and a multimodal DiT (MM-DiT). The MLLM handles visual–textual understanding, taking text, image, and video inputs and optionally producing text responses. The MM-DiT focuses on visual generation with two branches: one incorporates high-level semantic information from the MLLM, while the other integrates fine-grained reconstruction signals from a VAE. Specifically, we extract the last-layer hidden states of the MLLM, which encode rich semantic features of the multimodal input. These are aligned to the input space of the MM-DiT via a trainable connector and fed into its understanding stream. In parallel, visual signals are encoded by the VAE and passed into the MM-DiT generation stream to preserve fine details. This design enables strong semantic grounding together with high-fidelity visual detail, which is especially important for video editing and identity-preserving generation. We provide a model design analysis in subsection D.1.

### 2.2 UNIFYING MULTIPLE TASKS

We unify diverse multimodal tasks through natural language instructions, as illustrated in Figure 1. For text-to-video (T2V), the text input is processed by the MLLM, while the noisy video is fed into the MM-DiT. For image-to-video (I2V), both the image and text are processed by the MLLM, whereas the image and noisy video are provided to the MM-DiT. For in-context video generation (MultiID2V) and in-context video editing (ID-V2V), multiple visual conditions are often available, such as several reference images together with a reference video. Each visual signal is encoded with the VAE, padded to a uniform shape, concatenated along the temporal axis, and then processed with self-attention. Unlike prior approaches that introduce task-specific bias embeddings (Ye et al., 2025b) or context adapter modules (Jiang et al., 2025), we avoid task-specific customization. To help the MM-DiT distinguish between condition latents and noisy video latents, we apply 3D positional embeddings, which preserve the spatial indices across frames while incrementing only the temporal dimension. In practice, we find this strategy more effective than Qwen2-VL's MRoPE (Wang et al., 2024a), which offsets all axes whenever a new visual input is introduced.

### 2.3 GENERATION WITH THINKING

`UniVideo` leverages its MLLM branch to interpret unconventional or hand-crafted prompts, as illustrated in Figure 3 and Figure 6. For example, users may provide an input image with manual annotations, which the MLLM translates into a structured plan and dense prompt tokens that guide video generation. Unlike agent-based approaches that invoke multiple downstream generators, `UniVideo` offers a more simplified design: the MMDiT directly integrates embeddings from the dense prompt tokens produced by the MLLM.

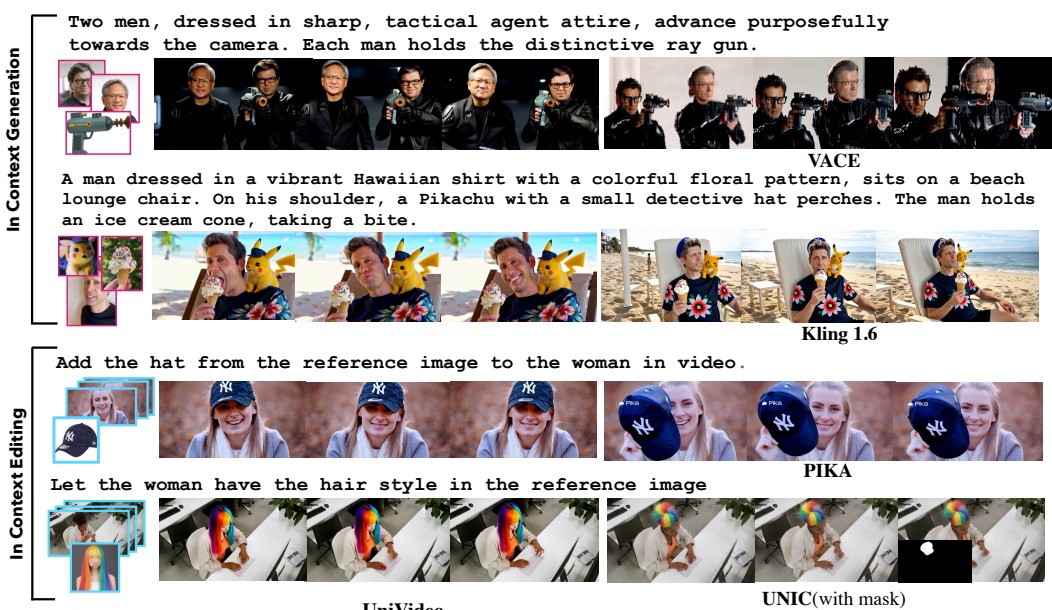

Figure 4: **Qualitative comparison** of `UniVideo` with SoTA Task Specific Experts on **In Context Generation** and **In Context Editing** tasks.

## 2.4 TRAINING STRATEGY

**Stage 1. Connector alignment between MLLM and MMDiT.** In this stage, we train only the MLP connector while keeping both the MLLM and MMDiT frozen. Training is performed on pretraining samples across text-to-image (T2I) and text-to-video (T2V) generation tasks, as well as an image-reconstruction task in which only images from the text-to-image dataset are fed into the MLLM and the MMDiT reconstructs the image using visual features from the MLLM. After this stage, `UniVideo` can generate images and videos conditioned on text or image inputs from the MLLM.

**Stage 2. Fine-tuning on T2I/T2V.** In this stage, we keep the MLLM frozen and fine-tune the connector and MMDiT on small-scale, high-quality T2I/T2V samples. After this stage, `UniVideo` achieves performance comparable to the MMDiT backbone that uses its own text encoder.

**Stage 3. Multi-task Training.** Finally, we extend training to include in-context generation (multi-ID-to-video), in-context video editing (modifying the input video based on a reference image, such as ID swapping, ID addition, ID deletion, or style transfer), image editing and image-to-video tasks, alongside the previous text-to-image (T2I) and text-to-video (T2V) tasks. We keep the MLLM frozen and only train the connector and MMDiT. This stage enables `UniVideo` to unify a broad range of video generation and editing tasks under multimodal instruction. Details of training setting is provided in Appendix Table 5.

## 3 EXPERIMENTS

In this section, we first describe the implementation details of `UniVideo` in subsection 3.1. Then, we present the main results in subsection 3.2. We conduct a comprehensive benchmark of `UniVideo` with SoTA methods across a broad spectrum of video understanding and generation tasks. Our results show that `UniVideo`'s strong unified capabilities across all settings. Next, we demonstrate the zero-shot generalization ability of `UniVideo` and analyze the visual prompt understanding ability in subsection 3.3. Finally, we validate the design choices of `UniVideo` through ablation studies in subsection 3.4. We discuss the design choices for aligning the MLLM and the Diffusion generator in Appendix subsection D.1.

Table 1: Quantitative comparison on **Visual Understanding and Video Generation**. Best results are shown in **bold**, and second-best are underlined. *Understanding results for `UniVideo` are reported using its MLLM backbone (Qwen-2.5VL-7B).

| Model | Understanding | | | Video Generation |
|---|---|---|---|---|
| | MMB | MMMU | MM-Vet | Vbench T2V |
| *Video Understanding Model* | | | | |
| LLaVA-1.5(Liu et al., 2024a) | 36.4 | **67.8** | 36.3 | × |
| LLaVA-NeXT(Liu et al., 2024b) | 79.3 | 51.1 | 57.4 | × |
| *Video Generation Model* | | | | |
| CogVideoX(T2V)(Yang et al., 2024) | × | × | × | 81.61 |
| I2VGen-XL(Zhang et al., 2023c) | × | × | × | × |
| HunyuanVideo(T2V)(Kong et al., 2024) | × | × | × | 83.24 |
| Step-Video-(T2V)(Ma et al., 2025a) | × | × | × | 81.83 |
| Wan2.1(T2V)(Wan et al., 2025) | × | × | × | **84.70** |
| *Unified Understanding & Generation Model* | | | | |
| Emu3 (Wang et al., 2024b) | 58.5 | 31.6 | 37.2 | 80.96 |
| TokenFlow-XL (Qu et al., 2025) | 76.8 | 43.2 | 48.2 | × |
| Janus (Wu et al., 2025b) | 69.4 | 30.5 | 34.3 | × |
| JanusFlow (Ma et al., 2025b) | 74.9 | 29.3 | 30.9 | × |
| OmniGen2 (Wu et al., 2025c) | 79.1 | 53.1 | 61.8 | × |
| Show-o (Xie et al., 2024) | - | 26.7 | - | × |
| BAGEL (Deng et al., 2025) | **85.0** | 55.3 | **67.2** | × |
| Show-o2 (Xie et al., 2025) | 79.3 | 48.9 | 56.6 | 81.34 |
| **`UniVideo`** * | 83.5 | 58.6 | 66.6 | 83.48 |

## 3.1 IMPLEMENTATION DETAILS

We adopt qwen2.5VL-7B (Bai et al., 2025b) as the MLLM backbone and HunyuanVideo-T2V-13B (Kong et al., 2024) as the MMDiT backbone. The original HunyuanVideo use two text encoders; we remove them and instead use qwen2.5VL as the unified multi-modal embedder. To align feature dimensions between qwen2.5VL and HunyuanVideo, we apply an MLP with a $4\times$ expansion. Additional details are provided in the Appendix.

## 3.2 MAIN RESULTS

### 3.2.1 VISUAL UNDERSTANDING AND GENERATION

`UniVideo`'s visual understanding is powered by a frozen pretrained MLLM. Freezing the MLLM preserves its strong native understanding ability and prevents performance degradation from joint training with generative tasks. As shown in Table 1, `UniVideo` achieves competitive scores of 83.5 on MMBench (Liu et al., 2024e), 58.6 on MMMU(Yue et al., 2024), and 66.6 on MM-Vet(Yu et al., 2023) for understanding tasks. At the same time, it retains strong generation ability, supporting both I2V and T2V within a single unified model. In contrast, baseline models rely on different variants for different tasks, whereas `UniVideo` reaches performance comparable to the HunyuanVideo backbone on the VBench(Huang et al., 2024) benchmarks.

### 3.2.2 IN-CONTEXT VIDEO GENERATION

**Benchmark:** Following FullDiT (Ju et al., 2025b) and OmniGen2 (Wu et al., 2025c), we construct a test set covering both single-ID and multi-ID video generation scenarios. In the single-ID setting, a subject may have multiple reference images (e.g., different viewpoints of a person or object). In the multi-ID setting, the references include 2–4 distinct identities. Details are provided in the Appendix.

**Metrics:** We conduct both human evaluations and automatic metric assessments. For human evaluation, we follow the protocols of Instruct-Imagen (Hu et al., 2024a) and OmniGen2 (Wu et al., 2025c) to perform a systematic study. Each sample is rated by at least three annotators on (i) subject

Table 2: Quantitative comparison on **In-Context Generation**. Human evaluation includes Subject Consistency (SC), Prompt Following (PF), and Overall Video Quality (VQ). Automatic metrics measure Smoothness and Aesthetics. Best results in **bold**, second-best underlined.

| Model | Single Reference | | | | | Multi Reference (≥2) | | | | |
|---|---|---|---|---|---|---|---|---|---|---|
| | SC↑ | PF↑ | VQ↑ | Smooth↑ | Aesth↑ | SC↑ | PF↑ | VQ↑ | Smooth↑ | Aesth↑ |
| VACE | 0.31 | 0.65 | 0.42 | 0.922 | 5.426 | 0.48 | 0.53 | 0.48 | 0.530 | 5.941 |
| Kling1.6 | 0.68 | **0.95** | 0.88 | 0.938 | **5.896** | 0.73 | 0.45 | **0.95** | 0.916 | 6.034 |
| Pika2.2 | 0.45 | 0.43 | 0.15 | 0.928 | 5.125 | 0.71 | 0.48 | 0.43 | 0.898 | 5.176 |
| **UniVideo** | **0.88** | 0.93 | **0.95** | **0.943** | 5.740 | **0.81** | **0.75** | 0.85 | **0.942** | **6.128** |

consistency (SC), (ii) prompt following (PF), and (iii) overall video quality (VQ). Scores in each category are drawn from $\{0, 0.5, 1\}$, where 0 indicates inconsistency or extremely poor quality, and 1 indicates full consistency or high quality. For automatic evaluation, we adopt three metrics from VBench (Huang et al., 2024): smoothness, and aesthetics.

**Baselines:** We compare `UniVideo` with the state-of-the-art open-source model VACE, given the scarcity of video models capable of in-context generation. We also include commercial baselines such as Pika2.2 and Kling1.6.

**Results:** Quantitative comparisons are presented in Table 2. `UniVideo` achieves superior or competitive performance across all metrics compared to the baselines. Additional results are shown in Figure 4, and more examples are available on our project website. Notably, baseline models often struggle with complex instructions involving multiple identities (e.g., when the number of reference images is 4), whereas `UniVideo` can accurately follow instructions while preserving identity.

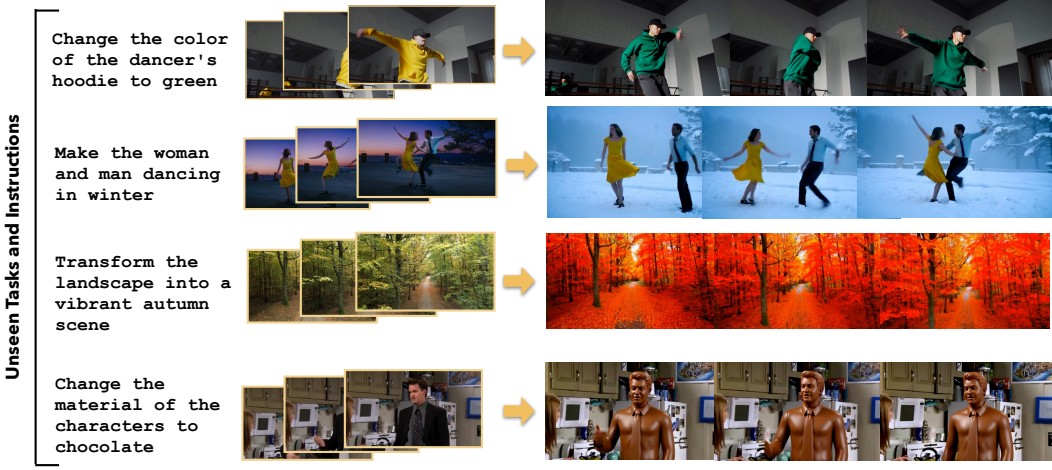

Figure 5: **Zero-Shot Generalization**. We demonstrate two type of generalization. (i) `UniVideo` was not trained on General Free-form Video Editing data. It transfers this ability from diverse image editing data to the video domain through joint training with in-context video generation and editing data (limited to ID deletion, swapping, addition, and stylization), enabling it to handle previously unseen video editing instructions. (ii) `UniVideo` can also generalize to novel task compositions, even though it was not explicitly trained on such compositions.

### 3.2.3 IN-CONTEXT VIDEO EDITING

**Benchmark:** Following UNIC (Ye et al., 2025b), we construct a test set covering four editing types: swap, delete, addition, and style transfer. Each example consists of a source video and a reference image, together with a natural language instruction. Further details are provided in the Appendix.

Table 3: Quantitative comparison with task-specific expert models on **In-Context Video Editing**. Our model is the **only mask-free approach**, capable of performing edits solely based on instructions without requiring explicit mask inputs to indicate editing regions. Despite this more challenging setting, it achieves superior or competitive performance across all metrics compared to state-of-the-art task-specific expert baselines. Best scores are shown in **bold**, and second-best are underlined.

| **In Context Insert** | | | | | |
|---|---|---|---|---|---|
| **Model** | Identity | | Alignment | Video Quality | |
| | CLIP-I↑ | DINO-I↑ | CLIP-score↑ | Smoothness↑ | Aesthetic↑ |
| VACE | 0.513 | 0.105 | 0.103 | 0.947 | 5.693 |
| UNIC | 0.598 | 0.245 | 0.216 | 0.961 | 5.627 |
| Kling1.6 | 0.632 | 0.287 | 0.246 | **0.993** | 5.798 |
| Pika2.2 | 0.692 | **0.399** | 0.253 | 0.951 | 5.591 |
| **UniVideo** *(Mask Free)* | **0.693** | 0.398 | **0.259** | 0.943 | **6.031** |
| **In Context Swap** | | | | | |
| **Model** | Identity | | Alignment | Video Quality | |
| | CLIP-I↑ | DINO-I↑ | CLIP-score↑ | Smoothness↑ | Aesthetic↑ |
| VACE | 0.703 | 0.391 | 0.218 | 0.960 | 5.961 |
| UNIC | 0.725 | 0.429 | 0.242 | 0.971 | 6.056 |
| Kling1.6 | 0.707 | **0.437** | 0.211 | **0.995** | 6.042 |
| Pika2.2 | 0.704 | 0.406 | 0.211 | 0.967 | 5.097 |
| AnyV2V | 0.605 | 0.229 | 0.218 | 0.917 | 4.842 |
| **UniVideo** *(Mask Free)* | **0.728** | 0.427 | **0.244** | 0.973 | **6.190** |
| **In Context Delete** | | | | | |
| **Model** | Video Reconstruction | | Alignment | Video Quality | |
| | PSNR↑ | RefVideo-CLIP↑ | CLIP-score↑ | Smoothness↑ | Aesthetic↑ |
| VACE | 20.601 | 0.874 | 0.206 | 0.968 | **5.637** |
| UNIC | 19.171 | 0.817 | **0.217** | 0.970 | 5.493 |
| Kling1.6 | 15.476 | 0.888 | 0.208 | **0.998** | 4.965 |
| AnyV2V | 19.504 | 0.869 | 0.205 | 0.964 | 5.325 |
| VideoPainter | **22.987** | **0.920** | 0.212 | 0.957 | 5.403 |
| **UniVideo** *(Mask Free)* | 17.980 | 0.888 | 0.214 | 0.971 | 5.498 |
| **In Context Stylization** | | | | | |
| **Model** | Style & Content | | Alignment | Video Quality | |
| | CSD-Score↑ | ArtFID↓ | CLIP-score↑ | Smoothness↑ | Aesthetic↑ |
| AnyV2V | 0.207 | 43.299 | 0.195 | 0.937 | 4.640 |
| StyleMaster | **0.306** | 38.213 | 0.188 | 0.952 | 5.121 |
| UNIC | 0.197 | **36.198** | 0.215 | 0.932 | 5.045 |
| **UniVideo** *(Mask Free)* | 0.228 | 37.877 | **0.226** | **0.963** | **6.281** |

**Metrics:** We adopt the evaluation protocol of UNIC (Ye et al., 2025b) and conduct automatic metric assessments. Specifically, we use CLIP-I and DINO-I to measure identity consistency, and CLIP-Score to measure prompt following.

**Baselines:** We compare UniVideo with state-of-the-art task-specific expert models, including UNIC, AnyV2V, and VideoPainter. We also evaluate against commercial models such as Pika2.2 and Kling1.6. **Note** that all baseline models require explicit mask inputs to localize editing regions and guide generation, whereas UniVideo operates without masks.

**Results:** Quantitative comparisons are presented in Table 3. Although UniVideo is evaluated under the more challenging mask-free setting, it still achieves superior or competitive performance across all metrics compared to the baselines. Additional results are shown in Figure 4, and further examples are provided on our project website. UniVideo can accurately follow instructions while preserving the identity of the reference images.

### 3.3 MODEL ANALYSIS

#### 3.3.1 ZERO SHOT GENERALIZATION

We observed two type of generalization ability of `UniVideo`. Although the training data of `UniVideo` does not include general free-form video editing tasks, it transfers this ability from diverse image editing data and in-context video editing data (limited to ID deletion, swapping, addition, and stylization) to the video domain, enabling it to handle free-form video editing instructions(e.g., changing material or environment). Surprisingly, we find that `UniVideo` can perform tasks such as changing materials of character. We also observe that `UniVideo` is capable of handling task compositions. It can combine in-context editing with style transfer, or perform multiple edits simultaneously (e.g., deleting one identity while adding another). Demonstrations in Figure 5.

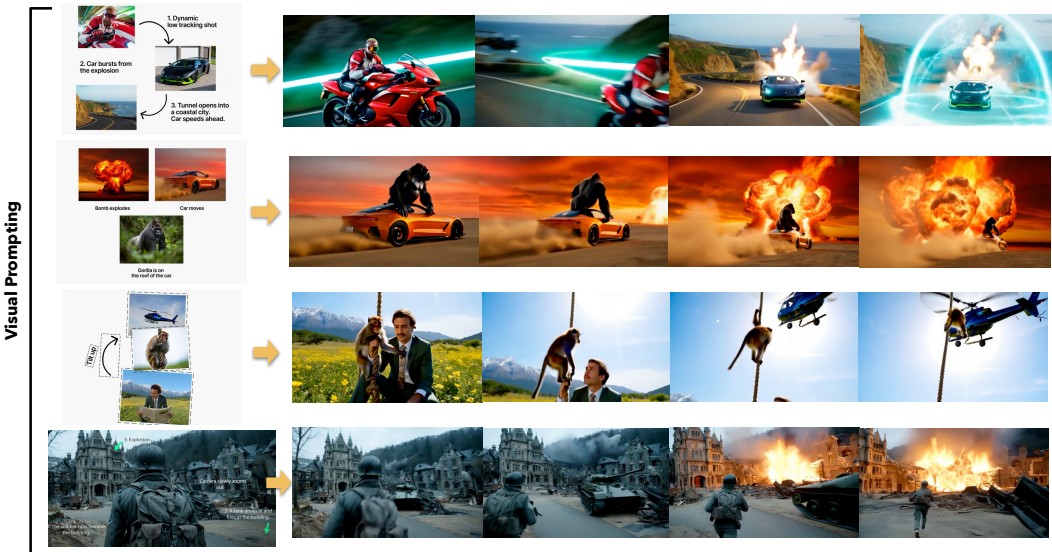

Figure 6: **Qualitative results of generation with thinking.** We illustrate two types of visual prompts: in the first three examples, annotations are drawn on a canvas, while in the last example, the annotation is drawn directly on an input image.

#### 3.3.2 GENERATION WITH THINKING

We demonstrate `UniVideo`'s *generation-with-thinking* capability in Figure 6, focusing on visual prompt understanding that requires structured multimodal reasoning. We consider two representative types of visual prompts. In the first setting, users draw reference images and story plans on a canvas. Here, the model can interpret the plan and generate corresponding videos. In the second setting, annotations are drawn directly on an input image, which the model treats as an I2V task; in this case, `UniVideo` can interpret the motion or new events described by the visual prompt. These results highlight the advantages of `UniVideo` in handling complex multimodal instructions. Although the qualitative results are obtained in a zero-shot setting, future end-to-end training on task-specific data may further improve performance.

### 3.4 ABLATION STUDY

Our ablation studies address the following two questions: (i) *Does multi-task learning enhance performance compared with single-task learning?* (ii) *Is our two-branch design effective? Specifically, should visual embeddings be streamed to both the MLLM and MMDiT branches?*

We conduct human evaluations on In-Context Video Editing and In-Context Video Generation, using the same evaluation protocol as in subsubsection 3.2.2. (i) To study multi-task learning, we compare `UniVideo` with a single-task baseline. The single-task baseline shares the same architecture as `UniVideo` but requires an independent model for each task and has access only to task-specific

Table 4: Ablation study across single-task model, `UniVideo`, and `UniVideo` w/o Visual for MMDiT across different In-Context tasks.

| | | Single-task | | | UniVideo | | | UniVideo w/o Visual for MMDiT | | |
|---|---|---|---|---|---|---|---|---|---|---|
| | | PF↑ | SC↑ | VQ↑ | PF↑ | SC↑ | VQ↑ | PF↑ | SC↑ | VQ↑ |
| In-context generation | singleid | 0.85 | 0.83 | 0.93 | 0.93 | 0.88 | 0.95 | 0.75 | 0.32 | 0.86 |
| | multiid | 0.75 | 0.79 | 0.73 | 0.75 | 0.81 | 0.85 | 0.81 | 0.23 | 0.83 |
| In-context editing | insert | 0.81 | 0.85 | 0.86 | 0.92 | 0.92 | 0.91 | 0.68 | 0.18 | 0.75 |
| | swap | 0.53 | 0.78 | 0.68 | 0.91 | 0.85 | 0.85 | 0.63 | 0.15 | 0.62 |
| | delete | 0.32 | 0.42 | 0.89 | 0.52 | 0.58 | 0.92 | 0.21 | 0.13 | 0.63 |
| | stylization | 0.56 | 0.43 | 0.63 | 0.79 | 0.64 | 0.64 | 0.86 | 0.11 | 0.57 |
| Average | | 0.64 | 0.67 | 0.79 | 0.80 | 0.78 | 0.85 | 0.66 | 0.18 | 0.71 |

data. Results in Table 4 demonstrate the effectiveness of multi-task learning, especially for the editing task, where `UniVideo` benefits from large-scale image editing data during joint learning. (ii) To evaluate the impact of streaming visual inputs, we compare `UniVideo` with a variant that share the same architecture: - **w/o visual for MMDiT:** visual inputs are fed only to the MLLM branch. As shown in Table 4, feeding visual inputs exclusively to the MLLM results in a dramatic drop in identity preservation.

## 4 RELATED WORK

**Unified Multimodal Understanding and Generation.** Recent progress in multimodal generation has been driven primarily by the text and image domains, spanning autoregressive modeling, diffusion–autoregression hybrids, and LLM-based regression approaches(Sun et al., 2024a; Team, 2024; Sun et al., 2024b; Wang et al., 2024b). While these advances demonstrate strong capabilities in images, unified approaches beyond the image domain remain limited. We instead present a unified video model. A full discussion of prior multimodal works is provided in Appendix subsection C.1.

**Image/Video Generation and Editing.** Diffusion models have achieved remarkable success in image and video synthesis(Rombach et al., 2022; Podell et al., 2023; Esser et al., 2024; Ramesh et al., 2021; Saharia et al., 2022) and unified image editing systems (Xiao et al., 2025; Tan et al., 2024; Chen et al., 2025d). In contrast, the video domain remains dominated by single-task frameworks. Attempts at unification(Ku et al., 2024; Ju et al., 2025b; Jiang et al., 2025) still require task-specific pipelines or modules. We bridge this gap by unifying diverse video tasks under a single framework. Extended related work in Appendix subsection C.2.

## 5 CONCLUSION

We introduce `UniVideo`, a unified multimodal generative model for video understanding, generation, and editing. By integrating an MLLM for semantic understanding with an MMDiT for generation, `UniVideo` combines strong multimodal reasoning with fine-grained visual consistency. It can interpret multimodal instructions and handle diverse tasks effectively. Our experiments show that `UniVideo` not only matches or outperforms task-specific baselines across text/image-to-video, video editing, and in-context generation, but also generalizes to unseen tasks and novel task compositions—capabilities that specialized pipelines struggle to achieve. Beyond robust performance, `UniVideo` can leverage the MLLM stream to interpret user intent from complex multimodal prompts such as visual prompts. Looking forward, `UniVideo` opens new directions for multimodal research, advancing us toward assistants that can naturally communicate through language, images, and video.

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

## A    APPENDIX

Appendix contains the following sections:

- Statement for Large Language Models
- Extended Related Work
- Additional Experiment and Analysis
- Training Details
- Limitation and Future Work
- Training Dataset Construction
- Evaluation Benchmark

## B    STATEMENT FOR LARGE LANGUAGE MODELS

We use large language models (LLMs) in this paper solely for grammar correction and text refinement. They are not employed for generating original content or contributing to the conceptual development of the ideas presented.

## C    EXTENDED RELATED WORK

### C.1    UNIFIED MULTIMODAL UNDERSTANDING AND GENERATION

Recent progress in multimodal generation has been driven primarily by the text and image domains. Autoregressive models such as LlamaGen, Chameleon, Emu2, and Emu3(Sun et al., 2024a; Team, 2024; Sun et al., 2024b; Wang et al., 2024b) adopt discrete token prediction. Hybrid approaches like Show-o, Transfusion, and DreamLLM (Xie et al., 2024; Zhou et al., 2024; Dong et al., 2023) integrate autoregression with diffusion for image synthesis. Regression- or instruction-tuning–based methods, including SEED-X, Janus, MetaMorph, Next-gpt and OmniGen2 (Ge et al., 2024; Wu et al., 2025b; Gupta et al., 2022; Wu et al., 2024; 2025c), adapt LLMs for image feature prediction and controllable generation. Efficiency-oriented designs such as LMFusion and MetaQueries (Shi et al., 2024a; Pan et al., 2025) freeze MLLMs and add lightweight modules or learnable queries, while large-scale pretraining efforts like Show-o2, BLIP3-o, MoGao, and BAGEL (Xie et al., 2025; Chen et al., 2025a; Liao et al., 2025; Deng et al., 2025; Liu et al., 2025c) demonstrate strong generalization on interleaved multimodal data. Despite these advances, most works remain centered on image understanding and generation. In contrast, we move beyond the image domain by presenting a unified video model. The most related works to ours are Omni-Video and UniVid (Tan et al., 2025; Luo et al., 2025), which primarily focus on the basic text-to-video generation task. However, these approaches do not investigate the potential benefits of a unified architecture—such as how unification can enhance compositional generalization in tasks like in-context editing and in-context generation. In contrast, our work explicitly demonstrates that a unified framework leads to stronger generalization to unseen tasks, highlighting the advantages of architectural unification across diverse understanding and generation scenarios.

### C.2    IMAGE/VIDEO GENERATION AND EDITING.

Diffusion models have achieved remarkable success in high-fidelity image synthesis, with systems like Stable Diffusion, DALL·E, and Imagen(Rombach et al., 2022; Podell et al., 2023; Esser et al., 2024; Ramesh et al., 2021; Saharia et al., 2022) establishing strong text-to-image capabilities and recent video diffusion models(Blattmann et al., 2023b; Polyak et al., 2024; Chen et al., 2025c; 2023; Yang et al., 2024; Blattmann et al., 2023a; Kong et al., 2024; Brooks et al., 2024; Ma et al., 2025a) enabling scalable video generation. To improve controllability, models including ControlNet, T2I-Adapter(Zhang et al., 2023b; Mou et al., 2024) introduce external condition modules, while editing frameworks like InstructPix2Pix, EMU-Edit (Brooks et al., 2023; Sheynin et al., 2024) support instruction-driven refinement. Recently, unified image generation has emerged, with OmniGen, OmniControl, and UniReal (Xiao et al., 2025; Tan et al., 2024; Chen et al., 2025d) expanding from

generation to reference-guided editing. General editing methods (Wei et al., 2024; Zhao et al., 2024; Liu et al., 2025b; Shi et al., 2024b; Zhang et al., 2023a) further highlight this trend. In contrast, the video domain remains dominated by single-task frameworks such as Video-P2P, MagicEdit, MotionCtrl (Liu et al., 2024c; Liew et al., 2023; Wang et al., 2024c; Liu et al., 2025a; Bai et al., 2025a; Huang et al., 2025a; 2026). Video Alchemist and Movie Weaver (Liang et al., 2025; Chen et al., 2024) are dedicated to in-context generation. Attempts at unification include AnyV2V(Ku et al., 2024), which requires task-specific pipelines, EditVerse(Ju et al., 2025a), which can not perform the visual understanding task. VACE(Jiang et al., 2025), which relies on heavy adapter designs, FullDiT(Ju et al., 2025b), which supports multi-condition video generation but lacks editing, and UNIC(Ye et al., 2025b), which unifies tasks but depends on task-specific condition bias, limiting scalability. Yet, compared to images, unified and flexible video generation and editing remains far less explored. Our work bridges this gap by unifying diverse video tasks under a multimodal instruction framework. We provide the model capabilities comparison in Table 8.

Table 5: Training hyperparameters across different stages. Stage 1: Connector alignment, Stage 2: Fine-tuning, Stage 3: Multi-task training.

| Hyperparameters | Stages | | |
| --- | --- | --- | --- |
| | Stage 1 (Connector Alignment) | Stage 2 (Fine-tuning) | Stage 3 (Multi-task) |
| Learning rate | $1 \times 10^{-4}$ | $2.0 \times 10^{-5}$ | $2.0 \times 10^{-5}$ |
| LR scheduler | Constant | Constant | Constant |
| Weight decay | 0.0 | 0.0 | 0.0 |
| Gradient norm clip | 1.0 | 1.0 | 1.0 |
| Optimizer | AdamW ($\beta_1 = 0.9, \beta_2 = 0.95, \epsilon = 1.0 \times 10^{-15}$) | | |
| Warm-up steps | 50 | 50 | 50 |
| Training steps | 15K | 5K | 15K |
| EMA ratio | - | 0.9999 | 0.9999 |
| Gen resolution (min, max) | (240, 480) | (480, 854) | (480, 854) |
| Gen frames (min, max) | (1, 1) | (1, 129) | (1, 129) |
| Und resolution (min, max) | (240, 480) | (480, 854) | (480, 854) |
| Und frames (min, max) | (1, 1) | (1, 4) | (1, 4) |
| Diffusion timestep shift | 5.0 | 5.0 | 5.0 |

# D  ADDITIONAL EXPERIMENT AND ANALYSIS

## D.1  MODEL DESIGN

Our model design study addresses the following question: *What is the most effective approach for aligning a pretrained MLLM with a diffusion generator during Stage 1 training?*

We investigate three design choices for aligning the pretrained MLLM with the diffusion generator in Stage 1. Throughout this stage, the MLLM remains frozen, while we vary the connector and DiT architectures across three variants as shown in  Figure 7.

**(a)** *Cross-attention DiT.* The first variant adopts a cross-attention–based DiT for text conditioning, where we replace its original text encoder with an MLP connector that projects the final hidden states from the MLLM into the DiT text embedding space. Both the MLP and DiT are trained.

**(b)** *Cross-attention DiT with Learnable query.* Building upon (a), we use a *learnable query* mechanism following Pan et al. (2025). Specifically, we extract the final hidden states of learnable queries from the MLLM, which are then passed through an MLP layer and used to replace the original text conditioning in the DiT's cross-attention module. We test two variants: (1) jointly training the

Table 6: Evaluation of **text-to-image generation** ability on Geneval (Ghosh et al., 2023) benchmark. † refer to the methods using LLM rewriter

| Type | Method | Single obj. | Two obj. | Counting | Colors | Position | Color attr. | Overall ↑ |
|---|---|---|---|---|---|---|---|---|
| Image Model | SDv2.1 | 0.98 | 0.51 | 0.44 | 0.85 | 0.07 | 0.17 | 0.50 |
| | SDXL | 0.98 | 0.74 | 0.39 | 0.85 | 0.15 | 0.23 | 0.55 |
| | IF-XL | 0.97 | 0.74 | 0.66 | 0.81 | 0.13 | 0.35 | 0.61 |
| | LUMINA-Next | 0.92 | 0.46 | 0.48 | 0.70 | 0.09 | 0.13 | 0.46 |
| | SD3-medium | 0.99 | 0.94 | 0.72 | 0.89 | 0.33 | 0.60 | 0.74 |
| | FLUX.1-dev | 0.99 | 0.81 | 0.79 | 0.74 | 0.20 | 0.47 | 0.67 |
| | NOVA | 0.99 | 0.91 | 0.62 | 0.85 | 0.33 | 0.56 | 0.71 |
| Unified Model | OmniGen | 0.98 | 0.84 | 0.66 | 0.74 | 0.40 | 0.43 | 0.68 |
| | TokenFlow-XL | 0.95 | 0.60 | 0.41 | 0.81 | 0.16 | 0.24 | 0.55 |
| | Janus | 0.97 | 0.68 | 0.30 | 0.84 | 0.46 | 0.42 | 0.61 |
| | Janus Pro | 0.99 | 0.89 | 0.59 | 0.90 | 0.79 | 0.66 | 0.80 |
| | Emu3-Gen† | 0.99 | 0.81 | 0.42 | 0.80 | 0.49 | 0.45 | 0.66 |
| | Show-o | 0.98 | 0.80 | 0.66 | 0.84 | 0.31 | 0.50 | 0.68 |
| | MetaQuery-XL† | – | – | – | – | – | – | 0.80 |
| | BLIP3-o† 4B | – | – | – | – | – | – | 0.81 |
| | BLIP3-o† 8B | – | – | – | – | – | – | 0.84 |
| | BAGEL | 0.99 | 0.94 | 0.81 | 0.88 | 0.64 | 0.63 | 0.82 |
| | UniWorld-V1 | 0.99 | 0.93 | 0.79 | 0.89 | 0.49 | 0.70 | 0.80 |
| | OmniGen2 | 1.00 | 0.95 | 0.64 | 0.88 | 0.55 | 0.76 | 0.80 |
| | **UniVideo** | 0.93 | 0.63 | 0.23 | 0.76 | 0.21 | 0.41 | 0.53 |
| | **UniVideo**† | 0.98 | 0.85 | 0.36 | 0.90 | 0.45 | 0.64 | 0.69 |

Table 7: Comparison results on ImgEdit-Bench (Ye et al., 2025a) and GEdit-Bench-EN (Liu et al., 2025b) (SC: Semantic Consistency, PQ: Perceptual Quality, O: Overall). "Overall" is calculated by averaging all scores across tasks. The best results are highlighted in **bold**, and the second-best results are underlined.

| Model | ImgEdit-Bench | | | | | | | | | | GEdit-Bench | | |
| | Add | Adj. | Ext. | Rep. | Rm. | Bg. | Sty. | Hyb. | Act. | Overall | G-SC | G-PQ | G-O |
|---|---|---|---|---|---|---|---|---|---|---|---|---|---|
| GPT-4o | 4.61 | 4.33 | 2.90 | 4.35 | 3.66 | 4.57 | 4.93 | 3.96 | 4.89 | 4.20 | 7.85 | 7.62 | 7.53 |
| MagicBrush | 2.84 | 1.58 | 1.51 | 1.97 | 1.58 | 1.75 | 2.38 | 1.62 | 1.22 | 1.90 | 4.68 | 5.66 | 4.52 |
| Instruct-P2P | 2.45 | 1.83 | 1.44 | 2.01 | 1.50 | 1.44 | 3.55 | 1.20 | 1.46 | 1.88 | 3.58 | 5.49 | 3.68 |
| AnyEdit | 3.18 | 2.95 | 1.88 | 2.47 | 2.23 | 2.24 | 2.85 | 1.56 | 2.65 | 2.45 | 3.18 | 5.82 | 3.21 |
| OmniGen | 3.47 | 3.04 | 1.71 | 2.94 | 2.43 | 3.21 | 4.19 | 2.24 | 3.38 | 2.96 | 5.96 | 5.89 | 5.06 |
| ICEdit | 3.58 | 3.39 | 1.73 | 3.15 | 2.93 | 3.08 | 3.84 | 2.04 | 3.68 | 3.05 | 5.11 | 6.85 | 4.84 |
| Step1X-Edit | 3.88 | 3.14 | 1.76 | 3.40 | 2.41 | 3.16 | 4.63 | 2.64 | 2.52 | 3.06 | 7.09 | 6.76 | **6.70** |
| BAGEL | 3.56 | 3.31 | 1.70 | 3.30 | 2.62 | 3.24 | 4.49 | 2.38 | 4.17 | 3.20 | **7.36** | 6.83 | 6.52 |
| UniWorld-V1 | 3.82 | 3.64 | **2.27** | 3.47 | 3.24 | 2.99 | 4.21 | 2.96 | 2.74 | 3.26 | 4.93 | **7.43** | 4.85 |
| OmniGen2 | 3.57 | 3.06 | 1.77 | 3.74 | 3.20 | 3.57 | 4.81 | 2.52 | 4.68 | 3.44 | 7.16 | 6.77 | 6.41 |
| **UniVideo** | **4.22** | **3.90** | 1.66 | **3.96** | **3.73** | **4.11** | 4.54 | **3.55** | **4.81** | **3.83** | 7.08 | 7.08 | 6.41 |

learnable queries, MLP layer, and DiT (as in Pan et al. (2025)); and (2) training only the learnable queries and MLP while keeping the DiT frozen.

**(c) UniVideo architecture.** The main difference in this variant lies in its use of MMDiT, which employs self-attention for joint text–video interaction instead of cross-attention. We replace MMDiT's original text encoder with an MLP connector that projects the final hidden states from the MLLM into the MMDiT's text embedding space. Only the MLP layer is trained, while both the MLLM and MMDiT remain frozen.

For the cross-attention variants, we use an internal model with an architecture similar to (Wan et al., 2025), originally based on a T5 text encoder(Raffel et al., 2020), which we replace with Qwen2.5-VL. For UniVideo, we follow the implementation details described in subsection 3.1. All variants are trained for 15K steps, and the qualitative results are presented in Figure 8.

Our findings show that the cross-attention variants require unfreezing the DiT generator to achieve effective alignment with the MLLM, as evidenced by the comparison between (b.2) and (b.1). Nevertheless, even after unfreezing, variants (a) and (b.1) exhibit limited text-following ability—particularly for compositional object prompts. In contrast, the UniVideo architecture achieves efficient and robust alignment by training only the MLP connector.

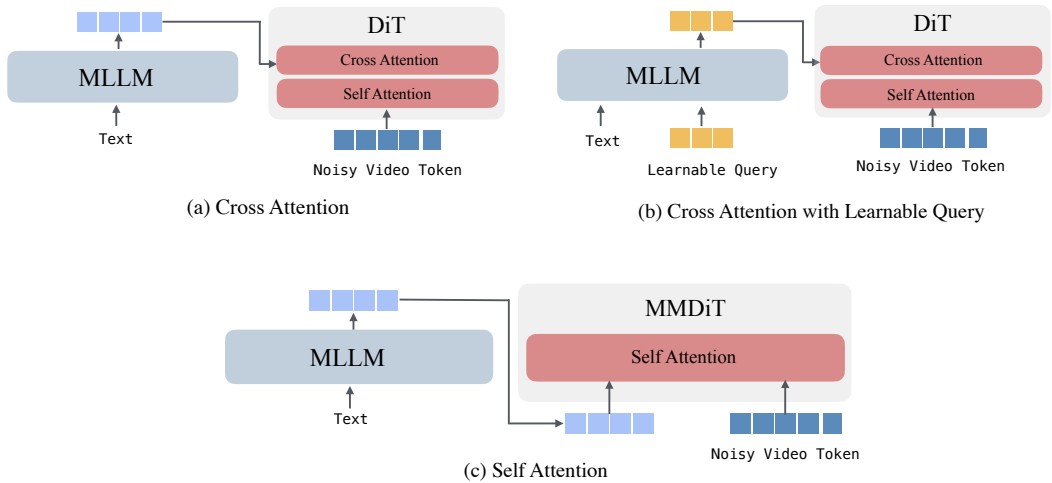

Figure 7: **Three design choices for aligning the MLLM with the diffusion generator in Stage 1 training.** We keep the MLLM fixed and vary the connector and DiT architecture across three variants: (a) the DiT uses cross-attention for text conditioning, where we replace its original text encoder with an MLP layer that aligns the final hidden states from the MLLM; (b) building upon (a), we introduce a learnable query design and extract the final hidden states from these learnable queries; and (c) our UniVideo architecture employs an MMDiT design that leverages self-attention for text conditioning.

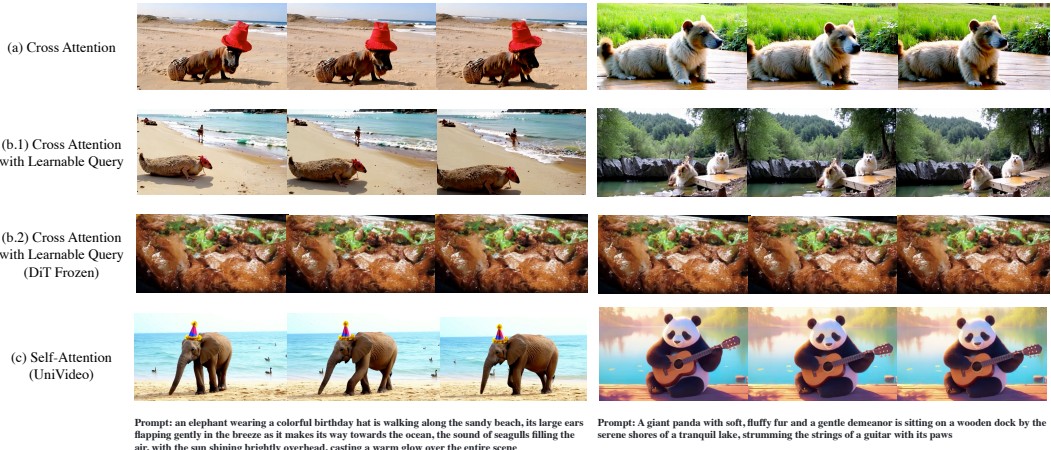

Figure 8: **Qualitative comparison of design choices for aligning the MLLM with the diffusion generator in Stage 1 training.** In all settings, the MLLM is kept frozen. (a) *Cross-Attention DiT:* we train the MLP connector and DiT; (b.1) *Cross-Attention DiT with Learnable Query:* following (Pan et al., 2025), we train the learnable query tokens, MLP connector, and DiT; (b.2) similar to (b.1), but the DiT is frozen while only the learnable query tokens and MLP connector are trained; (c) *UniVideo (MMDiT):* only the MLP connector is trained, with all other components frozen. All variants are trained for 15K steps. Among all variants, *UniVideo (MMDiT)* demonstrates the best prompt alignment.

*We study two `UniVideo` variants in Stage 3 training.* The MLLM is kept frozen, while we vary the connector design across two variants, as illustrated in Figure 9.

**(a) `UniVideo`(hidden).** In this variant, we extract the final-layer hidden states of all image, video, and text tokens produced by the MLLM. These token representations are used as inputs to the understanding branch of MMDiT. During training, only the MLP connector and the MMDiT are updated, while the MLLM remains frozen.

**(b) `UniVideo` (query).** This variant adopts a *learnable query* mechanism following Pan et al. (2025). Specifically, we extract the final hidden states of the learnable query tokens from the MLLM,

Figure 9: **Two design variants for multimodal conditioning in Stage 3 multi-task training.** We study two multimodal conditioning strategies: (a) extracting the final hidden states of image, video, and text tokens produced by the MLLM; and (b) adopting a learnable query design and using the final hidden states of the learnable queries together with the final hidden states of the text tokens.

together with the final hidden states of the text tokens. In this setting, we train the learnable queries, the MLP connector, and the MMDiT, while keeping the MLLM frozen.

`UniVideo` *(query)* can be more computationally efficient at inference time due to its fixed number of query tokens. By default, we use 512 learnable queries, which reduces the number of conditioning tokens compared to using all the multimodal hiddens from the MLLM. This efficiency gain is particularly beneficial for tasks where video inputs dominate the MMDiT conditioning, such as video editing.

During training, however, the `UniVideo` *(query)* variant requires backpropagation through the MLLM computation graph in order to optimize the learnable queries, which incurs additional memory overhead. In contrast, the `UniVideo` *(hidden)* variant does not require gradient flow through the MLLM and is therefore more memory-efficient during training.

### D.2 IMAGE GENERATION TASKS.

**Image Generation.** We evaluate `UniVideo` on text-to-image generation using the Geneval benchmark (Ghosh et al., 2023). Methods marked with † denote approaches that employ an LLM-based prompt rewriter. Despite being an image–video generalist model and using only a small proportion of T2I samples during Stage 3 training, `UniVideo` retains competitive text-to-image performance and outperforms many specialized text-to-image models and unified text–image multimodal models Table 6.

**Image Editing.** We further evaluate `UniVideo` on image editing tasks using ImgEdit-Bench (Ye et al., 2025a) and GEdit-Bench-EN (Liu et al., 2025b). We report Semantic Consistency (SC) as the primary evaluation metric. Despite being an image–video generalist model, `UniVideo` achieves strong editing performance and surpasses several specialized image editing models Table 7.

### D.3 VIDEO GENERATION TASKS.

For the **text-to-video generation task**, we use the prompt suite provided in VBench Huang et al. (2024), which contains 946 prompts covering 16 dimensions, including *subject consistency, background consistency, aesthetic quality, imaging quality, object class, multiple objects, color, spatial relationship, scene, temporal style, overall consistency, human action, temporal flickering, motion smoothness, dynamic degree, appearance style*. Following VBench, we obtain the semantic score and the quality score based on the scores on these dimensions. The results are shown in Table 9.

## E TRAINING DETAILS

We adopt qwen2.5VL-7B (Bai et al., 2025b) as the MLLM backbone and HunyuanVideo-T2V-13B (Kong et al., 2024) as the MMDiT backbone. The original HunyuanVideo also uses CLIP as its text encoder; we remove it and instead employ qwen2.5VL as the unified multimodal embedder. The released HunyuanVideo checkpoint is a CFG-distilled model, whose distillation embeddings we discard to simplify the training. To align feature dimensions between qwen2.5VL and Hunyuan-Video, we apply an MLP with a $4\times$ expansion. We report training configurations, hyperparameters in Table 5

## F   LIMITATION AND FUTURE WORK

Our model is trained on diverse tasks with multimodal instructions. While we do not observe task confusion, it sometimes fails to strictly follow editing instructions, occasionally over-editing unrelated regions. Due to backbone limitations, the model also struggles to fully preserve the motion of original videos, indicating the need for stronger video backbones. Moreover, although `UniVideo` generalizes to free-form video editing, its success rate remains lower than in image editing, underscoring the greater difficulty of video editing. Future work could explore large-scale video editing datasets and improved backbones for motion fidelity. Additionally, as `UniVideo` represents an assembled multimodal generative system capable of producing images, videos, and text, future work could aim to develop a native multimodal video model trained end-to-end.

## G   TRAINING DATASET CONSTRUCTION

This section details the construction of our datasets.

### G.1   ID-RELATED TASKS

For in-context video generation, which requires identity annotations, we follow the data creation pipeline of ConceptMaster (Huang et al., 2025b). We first extract keyframes from each video and then use Qwen2.5-VL-7B (Bai et al., 2025b) to identify the primary subjects in the video. The model is prompted to focus on semantically meaningful objects and ignore irrelevant background elements. Based on the subject tags generated by the Qwen2.5-VL-7B (Bai et al., 2025b), we obtain subject bounding boxes on the first frame with Grounding DINO (Liu et al., 2024d), We filter out videos with target areas that are either too small or too large. The lower bound is 10% of the frame and the upper bound is 60% of the frame. We then use apply SAM2 (Ravi et al., 2024) to obtain object segmentation masks from the source video. To further filter out object tracks that are not consistently visible (e.g., those that are too small in most frames or segmented unreliably), we compute a visibility consistency score. For each track, we count the number of frames in which the object's mask area exceeds a preset area threshold and divide this by the total number of frames in the track. Frames where the object is too small or poorly segmented do not contribute to the score. A higher score indicates that the subject remains clearly visible for most of the video. We discard tracks whose visibility consistency score falls below a predefined threshold. After this stage, we get sources videos and subject masks.

As demonstrated in Figure 10, to build in-context video tasks, we leverage an inpainter model.

For the object swap task, the inpainter is instructed to fill the masked region using the text tags predicted by Qwen2.5-VL (Bai et al., 2025b). To construct training pairs for this task, we use the inpainted video together with the subject image as the input, and the original video as the target.

For the object removal and addition tasks, we do not provide explicit textual instructions to the inpainter. Instead, the model fills the masked region based solely on the surrounding visual context, effectively removing the target object while preserving the background. For the addition task, we construct training pairs by using the inpainted video and the subject image as input, with the original video as the target. For the deletion task, we use the original video as the input and the inpainted video as the target.

To construct editing instructions for each pair of data, we employ Qwen2.5-VL-72B (Bai et al., 2025b) to generate precise editing instructions based on the first frame of the input video and and the first frame of the target video.

The inpainter is built on a 1B-parameter model with an architecture similar to Wan2.1 (Wan et al., 2025), which employs cross-attention modules for text conditioning and self-attention for visual tokens. We select and copy an interleaved half of the Transformer blocks from the original DiT to form the control net. While the original DiT processes noisy video tokens together with text tokens, the newly added control blocks operate on the masked video, the corresponding masks, and the text tokens. The output of each control block is injected back into the DiT as an additive control signal.

To train the video inpainter, we use the open source dataset VIVID-10M (Hu et al., 2024b), which provides source video and object mask for inpainter training.

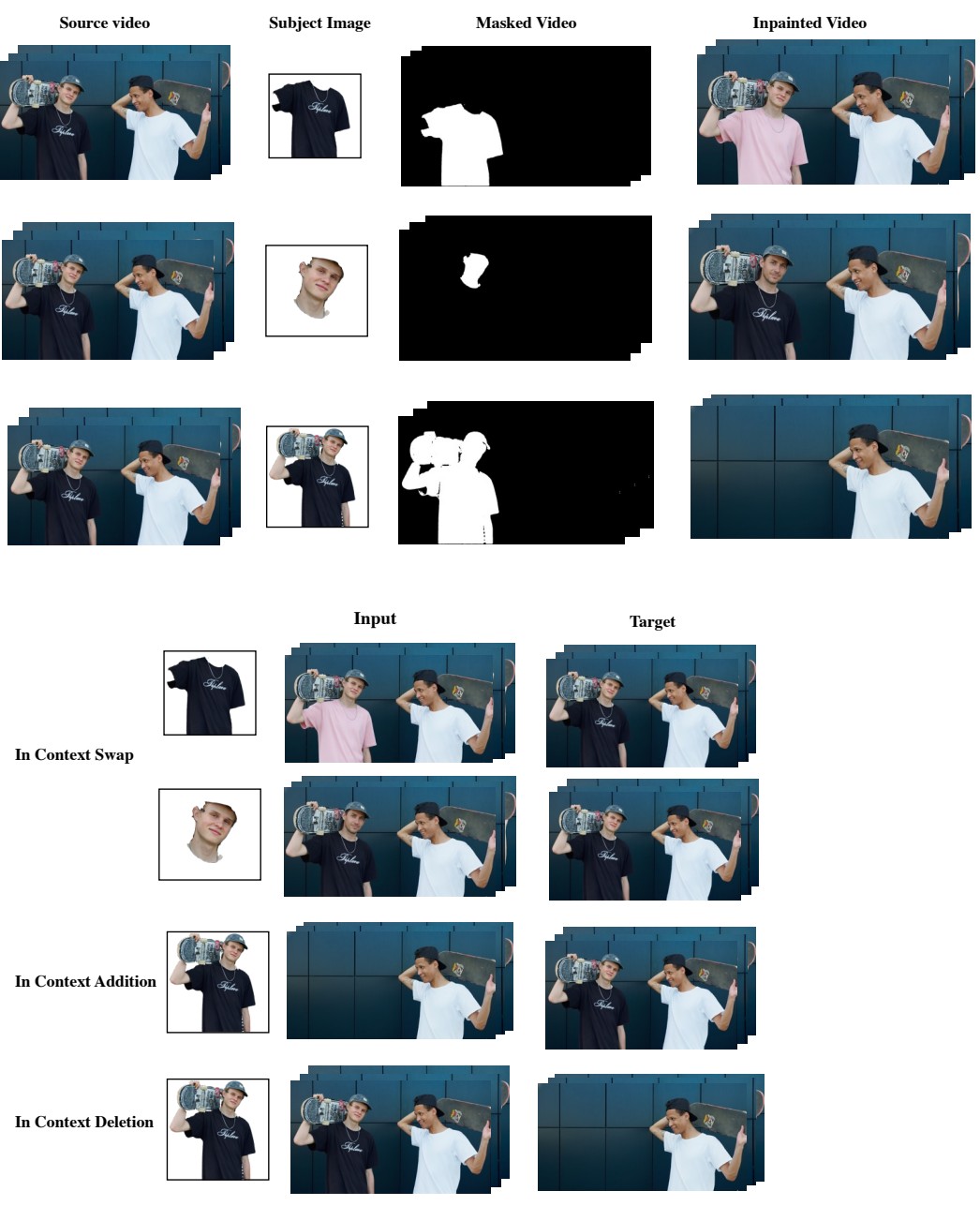

Figure 10: **In-Context task** dataset construction examples. The top section illustrates our pipeline: we first extract the subject image from the initial frame, then apply SAM2 (Ravi et al., 2024) to obtain video masks, and subsequently perform video inpainting based on these masks. The bottom section shows how we group the resulting images and videos into input–target pairs to form a dataset.

After constructing the dataset, we conduct a human filtering stage to ensure the final quality of all edited videos. Annotators are provided with both the source video and the edited video and evaluate each sample solely based on three criteria: *video quality*, *instruction following*, and *consistency with the source video(degree of overedit)*.

For object removal and addition tasks, a sample is accepted only if the edit satisfies all three dimensions: (1) high video quality, meaning the edited region is clear and artifact-free; (2) correct execution of the instruction, such as fully removing or appropriately adding the target object; and

**Annotator Instructions**
You are given two inputs:
– Source video
– Edited video

**A sample should be accepted only if it satisfies all three dimensions:**

**1. Video Quality**

- The edited region is clear, stable, and free of severe blur.
- No obvious artifacts such as texture duplication, holes, melting shapes, or structural collapse.
- Motion is temporally consistent with no strong flicker or jitter.

**2. Instruction Following**

- The edit correctly follows the given instruction (e.g., object removal, addition, or swap).

**3. Consistency With the Source Video**

- No unintended changes or *over-editing* outside the target region.
- The edited content matches the original motion, lighting, and scene dynamics across frames.

Figure 11: Annotator instruction used for human filtering of in context task video data.

(3) consistency with the original video, ensuring natural backgrounds and no over-editing beyond the target region. Any sample exhibiting artifacts, partial edits, or temporal flicker is rejected.

For object swap tasks, annotators apply the same three metrics. A sample is accepted only if (1) the edited content is visually stable and free of distortions, (2) the swap operation correctly follows the instruction, and (3) the resulting video remains consistent with the original motion, lighting, and scene dynamics. Samples containing structural distortions, unnatural textures, or temporal inconsistency are rejected. Identity verification is unnecessary, as the source video already defines the intended target appearance.

## G.2 STYLIZATION

Following UNIC (Ye et al., 2025b), Text-to-Video (T2V) models are capable of generating stylized videos with high visual quality and strong fidelity to a given reference style image. Instead of directly stylizing an existing real video, we leverage this capability to first produce a high-quality stylized video using a T2V model. We then convert this stylized video into a realistic counterpart using a stylized-to-real ControlNet Video DiT model.

The input to the ControlNet is a *gray tile signal*. Specifically, we downsample the video spatially by a factor of 8 and then upsample it by the same factor to remove high-frequency details, producing a low-fidelity tile image. We further discard the color information by converting this tile image into grayscale. This results in a structural guidance signal that preserves spatial layout while suppressing style and texture.

Similar to StyleMaster (Ye et al., 2025c), the ControlNet is built on a 1B-parameter DiT architecture similar to Wan2.1 (Wan et al., 2025), which combines cross-attention for text conditioning with self-attention over visual tokens. We construct the ControlNet by copying an interleaved half of the Transformer blocks from the original DiT. While the original DiT processes noisy video tokens alongside text tokens, the ControlNet blocks operate on the gray tile signal together with the text tokens. The output of each ControlNet block is injected back into the DiT through additive residual connections.

Table 8: Model capabilities across understanding, generation, editing, and in-context generation. ✓indicates support; ✗indicates not supported. The last row is highlighted.

| Model | Understanding | Image Gen. | Video Gen. | Image Edit. | Video Edit. | In-context Video Gen. |
|-------|:---:|:---:|:---:|:---:|:---:|:---:|
| LLaVA-1.5 | ✓ | ✗ | ✗ | ✗ | ✗ | ✗ |
| SD3-medium | ✗ | ✓ | ✗ | ✗ | ✗ | ✗ |
| FLUX.1-dev | ✗ | ✓ | ✗ | ✗ | ✗ | ✗ |
| QwenImage | ✓ | ✓ | ✗ | ✓ | ✗ | ✗ |
| HunyuanVideo | ✗ | ✓ | ✗ | ✗ | ✗ | ✗ |
| Show-o | ✓ | ✓ | ✗ | ✗ | ✗ | ✗ |
| Janus-Pro | ✓ | ✓ | ✗ | ✓ | ✗ | ✗ |
| Emu3 | ✓ | ✓ | ✗ | ✓ | ✗ | ✗ |
| BLIP3-o | ✓ | ✓ | ✗ | ✗ | ✗ | ✗ |
| BAGEL | ✓ | ✓ | ✗ | ✓ | ✗ | ✗ |
| OmniGen2 | ✓ | ✓ | ✗ | ✗ | ✗ | ✗ |
| VACE | ✗ | ✓ | ✓ | ✗ | ✗ | ✓ |
| **UniVideo** | ✓ | ✓ | ✓ | ✓ | ✓ | ✓ |

We train the stylized-to-real ControlNet using 10K video pairs in which both the input and target videos are real. During training, the model therefore learns a real-to-real reconstruction task. Since the control signal (the gray tile) preserves only coarse spatial structure while discarding color, details, and style, the model learns to generate realistic content guided only by spatial layout. At inference time, the model can effectively perform stylized-to-real mapping because the stylized input video is also converted into a gray-tile signal, which contains only spatial layout information and thus matches the training distribution.

### G.3 Image Editing, Text-to-Video and Text-to-Image

We leverage state-of-the-art image-editing models such as FLUX.1 Kontext (Labs et al., 2025) to construct a diverse collection of edited images. We further incorporate high-quality open-source datasets, including OmniEdit (Wei et al., 2024), ImgEdit (Ye et al., 2025a), and ShareGPT-4o-Image (Chen et al., 2025b). Following OmniEdit, we apply an additional VLM-based filtering stage on the curated image-editing dataset. Each (source, edited) pair is evaluated using Qwen2.5-VL, which assigns 0–10 scores along three core dimensions:

- **Image Quality:** the edited region must be sharp and visually stable, with no artifacts such as duplicated textures, holes, melting shapes, unnatural boundaries, or structural distortions.

- **Instruction Following:** the edit must correctly execute the given instruction (e.g., object removal, addition, or swap), without partial or incorrect modifications.

- **Consistency With the Source Image(degree of overedit):** no unintended changes or over-editing may occur outside the target region, and the edited content must remain coherent with the original scene's lighting, colors, and geometry.

Samples falling below threshold on any dimension are discarded. After filtering, we retain approximately 500K high-quality edited samples.

For text-to-image and text-to-video generation tasks, we utilize additional internal datasets.

## H Evaluation Benchmark

### H.1 In-Context Video Generation

For the in-context video generation, we construct a test set consisting of 20 cases, evenly split between single-ID and multi-ID scenarios. For each case, we collect ID images and carefully design prompts to ensure reasonable evaluation. As shown in Fig. 12, we build an ID pool with diverse images, ranging from cartoons to real-world subjects, including humans, animals, and common objects. We then select ID images from this pool and design appropriate prompts for them.

Table 9: **T2V Generation Performance on VBench**

| Model Name | Quality | Semantic | Subject | Background | Temporal | Motion | Dynamic | Aesthetic | Imaging |
|---|---|---|---|---|---|---|---|---|---|
| EasyAnimateV5.1 | 85.03 | 77.01 | 98.00 | 97.41 | 99.19 | 98.02 | 57.15 | 69.48 | 68.61 |
| MiniMax-Video-01 | 84.85 | 77.65 | 97.51 | 97.05 | 99.10 | 99.22 | 64.91 | 63.03 | 67.17 |
| Kling 1.6 | 85.00 | 76.99 | 97.40 | 96.84 | 98.64 | 99.13 | 62.22 | 64.81 | 69.70 |
| Wan2.1-T2V-1.3B | 85.23 | 75.65 | 97.56 | 97.93 | 99.55 | 98.52 | 65.19 | 65.46 | 67.01 |
| HunyuanVideo | 85.09 | 75.82 | 97.37 | 97.76 | 99.44 | 98.99 | 70.83 | 60.36 | 67.56 |
| Gen-3 | 84.11 | 75.17 | 97.10 | 96.62 | 98.61 | 99.23 | 60.14 | 63.34 | 66.82 |
| Vchitect-2.0 (VEnhancer) | 83.54 | 77.06 | 96.83 | 96.66 | 98.57 | 98.98 | 63.89 | 60.41 | 65.35 |
| TUNA | 84.32 | 83.04 | 95.99 | 96.72 | 98.02 | 98.33 | 69.39 | 65.88 | 66.83 |
| CogVideoX1.5-5B | 82.78 | 79.76 | 96.87 | 97.35 | 98.88 | 98.31 | 50.93 | 62.79 | 65.02 |
| Ours | 84.36 | 79.96 | 96.57 | 96.66 | 99.29 | 99.15 | 49.72 | 68.68 | 67.32 |

| Model Name | Object | Multi-Obj | Action | Color | Spatial | Scene | Appearance | Temporal | Overall |
|---|---|---|---|---|---|---|---|---|---|
| EasyAnimateV5.1 | 89.57 | 66.85 | 95.60 | 77.86 | 76.11 | 54.31 | 23.06 | 24.61 | 26.47 |
| MiniMax-Video-01 | 87.83 | 76.04 | 92.40 | 90.36 | 75.50 | 50.68 | 20.06 | 25.63 | 27.10 |
| Kling 1.6 | 93.34 | 63.99 | 96.20 | 81.26 | 79.08 | 55.57 | 20.75 | 24.51 | 26.04 |
| Wan2.1-T2V-1.3B | 88.81 | 74.83 | 94.00 | 89.20 | 73.04 | 41.96 | 21.81 | 23.13 | 25.50 |
| HunyuanVideo | 86.10 | 68.35 | 94.40 | 91.00 | 68.68 | 53.88 | 19.80 | 25.89 | 26.44 |
| Gen-3 | 87.81 | 53.64 | 96.40 | 85.95 | 65.09 | 54.57 | 24.31 | 24.71 | 26.11 |
| Vchitect-2.0 (VEnhancer) | 86.61 | 68.84 | 97.20 | 87.04 | 57.55 | 56.57 | 25.01 | 23.73 | 27.57 |
| TUNA | 95.41 | 92.31 | 97.50 | 87.67 | 78.12 | 58.59 | 23.18 | 24.68 | 27.71 |
| CogVideoX1.5-5B | 87.47 | 69.65 | 97.20 | 87.55 | 80.25 | 52.91 | 24.89 | 25.19 | 27.30 |
| Ours | 94.59 | 82.06 | 97.40 | 82.83 | 76.43 | 48.74 | 24.92 | 25.06 | 26.38 |

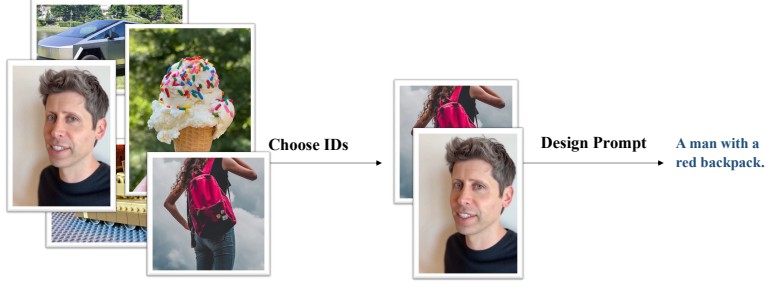

Figure 12: **Construction pipeline of in-context video generation test set.**

The single-ID examples are shown in Fig. 13. The single ID can have either one ID image, as shown by the cat example, or multiple shots of the same ID, as demonstrated by the human example.

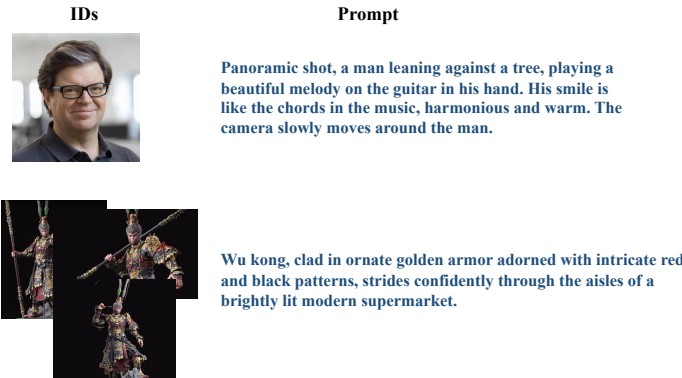

Figure 13: **Example of single-ID test case in in-context video generation test set.**

As shown in Fig. 14, in the multiple-ID scenarios, the number of IDs in a case ranges from 2 to 4, with larger numbers leading to higher difficulty. Our prompts focus on the interaction between these ID images and describe the relationships among them. For example, in the first case, the prompt describes a woman sitting on the sofa beside the bag, which connects the woman, sofa, and bag provided in the ID images. In the second case, the relationship between the two characters is described as Psyduck riding Pikachu.

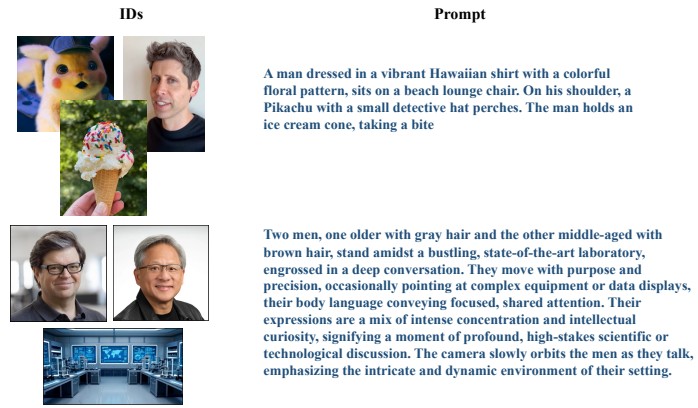

Figure 14: **Example of multi-ID test case in in-context video generation test set.**

## H.2 In-Context Video Editing

For the in-context video editing, we evaluate on the UNICBench Ye et al. (2025b) across four tasks: ID Insertion, ID Swap, ID Deletion, and Stylization. Since our setting differs from other video editing models (which may require masks to indicate the edited area, while ours uses instructions instead), we demonstrate in detail how we derive our inputs from the existing video editing benchmark.

First, as shown in Fig. 15, for ID insertion, the elements in UNICBench consist of a reference video, reference ID, and a caption for the target video. The goal of ID insertion is to naturally integrate new objects or elements from the reference ID into the target video. Here we replace the caption with a more direct instruction.

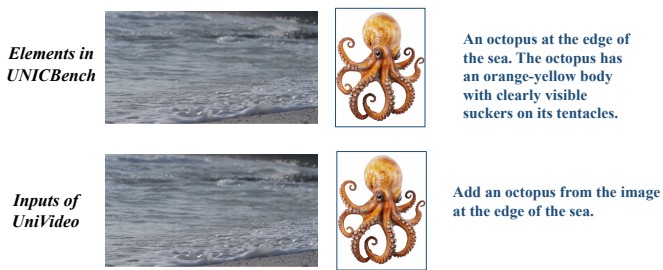

Figure 15: **Example of ID insertion test case.**

For ID swap, the elements in UNICBench consist of a reference video, mask, reference ID, and a caption for the target video. The goal of ID swap is to replace specific elements in the target video with corresponding elements from the reference ID while preserving the original video's context and motion. In our setting, we don't need a mask to indicate the editing area; instead, we use a more

convenient instruction-based approach. For example, in Fig. 16, we simply use the instruction "Use the man's face in the reference image to replace the man's face in the video."

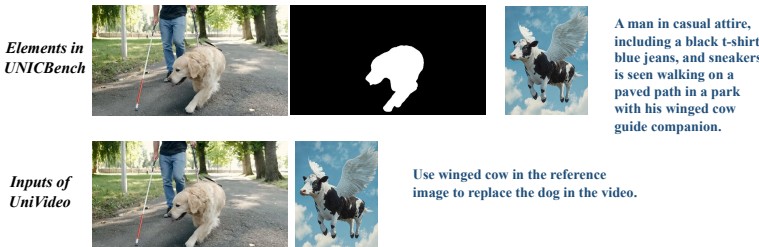

Figure 16: **Example of ID swap test case.**

For ID deletion, UNICBench provides a reference video, mask, and a caption for the target video. ID deletion aims to naturally remove specified objects or elements from the video while maintaining visual consistency and filling the removed areas with appropriate background content. While current video editing methods use masks to specify the object for removal, our approach simplifies this through text instructions. As demonstrated in Fig. 17, we use straightforward prompts such as "Delete the computer in the video."

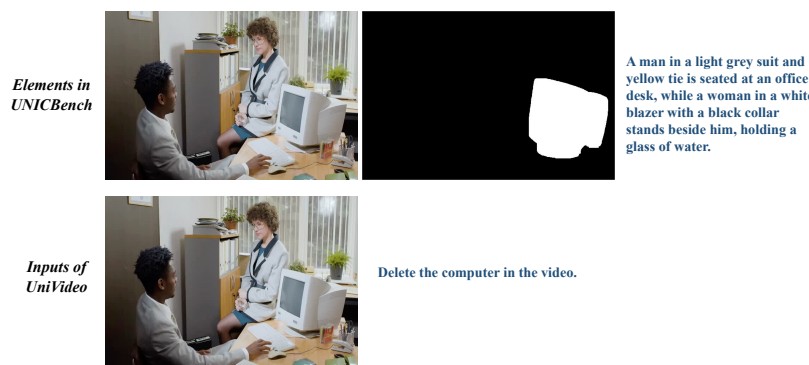

Figure 17: **Example of ID deletion test case.**

For stylization, the existing elements in UNICBench include a style reference image, target caption, and reference video. The purpose of stylization is to transform the visual appearance of the target video to match the artistic style of the reference image while preserving the original video's content and motion dynamics. We standardize the instruction format to "Transform the video into the style of the reference image," as shown in Fig. 18.

*Elements in
UNICBench*

*Inputs of
UniVideo*

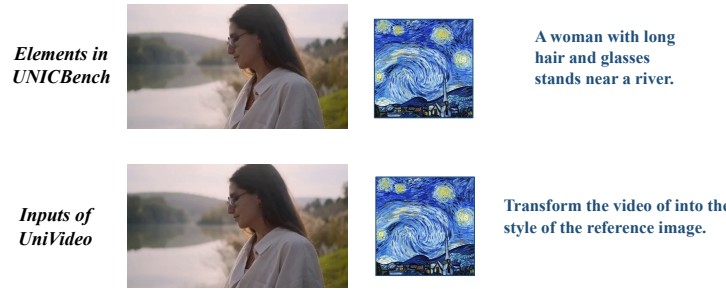

A woman with long
hair and glasses
stands near a river.

Transform the video of into the
style of the reference image.

Figure 18: **Example of stylization test case.**

