# OpenReview forum: "UniVideo: Unified Understanding, Generation, and Editing for Videos"
_ICLR.cc/2026/Conference — ICLR 2026 Poster_

### Official Review · Reviewer_dJ5L · 2025-10-19

**Soundness:** 4
**Presentation:** 4
**Contribution:** 3
**Rating:** 6
**Confidence:** 5

**Summary:**

This paper introduces VOGUE, a unified framework for video understanding, generating, and editing from a single multimodal prompt. To achieve this, VOGUE includes a MLLM for complex semantic and instruction understanding, which is followed by a MMDiT for high-fidelity video generation. The authors demonstrate that VOGUE is able to understand complex multimodal instructions and perform diverse tasks such as T2V, I2V, in-context generation, in-context editing, and even zero-shot generalization. The experiments also show that VOGUE outperforms SOTA task-specific methods on multiple benchmarks like VBench, UNICBench, and human evaluations.

**Strengths:**

- The most impressive strength of the work is the wide coverage on diverse tasks, including video understanding, T2V, I2V, in-context video generation, and in-context video editing, under a single framework. As far as I understand, VOGUE is the first video generation model that achieves this level of task unification which can substantially enhance the community interest.

- I also like the experiments where the authors show the model can achieve some zero-shot generation tasks which are not explicitly covered during training. For example, it can transfer the editing abilities to the unseen tasks composition (such as combining style transfer with object deletion). This highlights the advantages to leverage the powerful and frozen MLLM.

- The experiments include the comparisons with existing models on eight tasks. It is also impressive that such a general-purposed model can achieve a similar or comparable performance with the tasks-specific models.

- The authors also provide thorough ablation studies, including 1) the necessity of multi-task learning (compared to single-task learning) and 2) dual-stream model architecture, which are insightful.

- The authors promise that they will release the model checkpoint and code, which can significantly increase the reproducibility and also benefit the further research in unified video generation and editing.

- The paper is well written and easy to follow. The figures are well-plotted and informative which can make readers quickly understand the core ideas.

**Weaknesses:**

- My major concern is the incremental technical contribution. The dual-stream architecture that separately processes multimodal instructions and visual inputs has been explored in some existing models such as FLUX, which also employs two streams to process text and image inputs. Also, using MLLM embeddings to condition the diffusion process has also been explored in Qwen-Image. Therefore, while the model design is effective, it feels more like an integration of existing ideas and does not look insightful to me.
- It is unclear why the dual-stream architecture improves generalization compared to single-stream model design. For example, models like FullDiT [1] and Qwen-Image achieve great performance using full self-attention without separating understanding and generation streams. Could the authors provide the motivation behind this architecture design?
- The paper provides limited information about the construction of the training dataset. I hypothesize that learning such a general-purpose model requires high-quality, large-scale, and carefully-curated datasets for training. However, the authors only give a brief summary in Appendix F without describing more details such as the data filtering or annotation. It raises concerns about how the model learns to cover complex tasks such as visual prompt understanding.
- Following the above concerns, while Figure 6 shows several types of visual prompts, it is unclear how the model handles out-of-distribution or ambiguous prompts.
- There are some missing citations and comparisons with existing multi-subject video personalization models, such as Video Alchemist [2] and Movie Weaver [3], which can also be formulated as in-context generation.

[1] "FullDiT: Multi-Task Video Generative Foundation Model with Full Attention", ICCV 2025

[2] "Multi-subject Open-set Personalization in Video Generation", CVPR 2025

[3] "Movie Weaver: Tuning-Free Multi-Concept Video Personalization with Anchored Prompts", CVPR 2025

**Questions:**

- Could the authors explain the theoretical motivation for the proposed dual-stream architecture and include ablation studies to verify the design choice of separating the understanding and generation streams?
- Could the authors elaborate on the dataset curation pipeline?
- For the visual prompting task, could the authors describe whether the model can handle OOD or ambiguous prompts, and provide some failure cases to help check the generality of this task?

---

> ### Author Response · Authors · 2025-11-25
> **Response to Reviewer dJ5L (1/n)**
>
> Dear reviewer `dJ5L`,
>
> We sincerely thank the reviewer for the encouraging and insightful feedback, and for recognizing that VOGUE among one of the first unified video models. We are encouraged that the reviewer finds our emergent zero-shot generation abilities impressive and believes our work can substantially enhance community interest in unified video generation. We also appreciate the recognition that our ablation studies on multi-task learning and the dual-stream architecture provide valuable insights.
>
> >**W1: Model Design–Existing Work Comparison**
>
> **Answer:** We thank the reviewer for this thoughtful comment.
>
> **We want to clarify that the Qwen-Image preprint was released on `August 4, 2025`, while our work was submitted on `September 19, 2025`; thus, the two works represent concurrently developed efforts(within 2 months). Qwen-Image explores a related design idea in the image editing setting; our model targets the video domain and a multitask unified setting.**
> With this context, we would like to further clarify that our work provides distinct insights and contributions in the following aspects:
>
> (1) Studies such as Qwen-Image [2] directly apply MLLM embeddings, but **do not systematically examine whether an MLLM is necessary compared to a standard text encoder**. In contrast, we explicitly compare against a baseline that uses the text encoder and demonstrate consistent improvements, particularly in tasks requiring visual grounding and complex multimodal understanding.
>
> (2) How to effectively align an MLLM with a diffusion generator remains unclear in existing works. **Existing works do not systematically analyze different alignment strategies**. In our work, we conduct a detailed study of multiple design variants and show that the VOGUE architecture is the most effective among them, which provides new technical insight.
>
> (3) Existing approaches such as FLUX [1] and Qwen-Image [2] primarily focus on **single-domain, task-specific or purely image-generation settings**. In contrast, VOGUE’s design enables unified modelling of diverse tasks (visual understanding, T2V, I2V, in-context generation and editing, instruction editing, visual prompt understanding) and is among the first to study a unified video model, which has not been explored in prior work.
>
> Below, we explain each point in detail
>
> **First, the necessity of MLLM embeddings.**
> We include in Appendix G.2 a direct comparison between:
> VOGUE, and
> VOGUE without MLLM,
> using the same dataset and training setup.
> We also provide qualitative comparisons on the [anonymous project website](https://anonymous-submission-rebuttal.github.io/vogue/).
> Our analysis shows that the MLLM is particularly important for tasks requiring strong visual grounding. For example, in in-context generation, when the reference image is not a close-up shot of a single object and instead contains multiple objects, the model must correctly ground the instruction to the appropriate region or entity. Models using only the original text encoder often fail in such cases.
> Additionally, in editing tasks that require fine-grained grounding—such as deleting a small object at the border of the frame (e.g., a clock on the wall), or swapping an object at the edge of the video (e.g., a paper bag on the floor), or tasks requiring prior visual knowledge (e.g., replacing an object with Pikachu). The VOGUE w/o MLLM baseline often fails to follow these instructions, whereas VOGUE succeeds.
>
> These observations go beyond existing works (e.g., FLUX[1], Qwen-Image[2]), which do not study such visual grounding behavior nor systematically evaluate MLLM vs. non-MLLM conditioning. Our study provides the community with novel technical insight.
>
>
> **Second, aligning an MLLM with a video diffusion generator is nontrivial, and our study provides new technical insights.**
> In the updated submission, we include our early model design study in Appendix G.2, where we systematically study how to effectively align a pretrained MLLM with a video diffusion generator. We exhaustively explored several alignment strategies variants widely used in the image domain:
>
> (1) using an MLP to align a pretrained MLLM with a cross-attention DiT, training only the MLP;
>
> (2) using learnable queries to align with a cross-attention DiT, training only the queries;
>
> (3) using learnable queries with a cross-attention DiT, training both the queries and the DiT;
>
> (4) our final VOGUE architecture: using an MLP to align an MLLM with a self-attention MM-DiT, where text–video interaction occurs through self-attention rather than cross-attention.
>
> Our findings show that the proposed VOGUE design is the most effective among these options: under the same training steps, VOGUE achieves the strongest performance with minimal trainable parameters.
>
> These findings provide new, practical, and actionable technical insights for building unified video models during post-training—insights not explored in works such as FLUX[1] or Qwen-Image[2].

---

> ### Author Response · Authors · 2025-11-25
> **Response to Reviewer dJ5L (2/n)**
>
> **Third: Existing works focus on single-domain, task-specific designs, whereas VOGUE enables unified modeling with emergent capabilities, providing a novel conceptual advance**
> Existing works such as FLUX are dedicated to image generation, and Qwen-Image[2] / Qwen-Image-Edit[2] target image generation and editing, respectively. These models are designed as task-specific expert models.
> In contrast, our study aims at building a unified model for diverse tasks. Most importantly. Our work provides a conceptual advance by answering why we should build unified models.
> We directly demonstrate the benefits of unified modelling through two emergent synergies:
>
> (a)First, VOGUE enables two forms of generalization. Figure 5 and demos on the website show VOGUE
> generalize to task composition, such as combining editing with style transfer, by integrating multiple capabilities within a single instruction.
> Second, even without explicit training on free-form video editing, VOGUE transfers its editing capability from large-scale image editing data to this setting and generalizes to unseen instructions, such as changing the environment or changing materials within a video.
>
> (b)Second, our ablation in Table 5 further shows that multi-task training improves performance on every task compared to single-task experts. Specifically, we compare VOGUE with single-task baselines. Single-task baselines share the same architecture as VOGUE but require an independent model for each task and have access only to task-specific data. VOGUE consistently outperforms the baseline on all tasks, which highlights the importance of joint task learning.
>
> These insightful findings and benefits are not achievable without the architectural design choices made in VOGUE. Therefore, our contributions provide both conceptual and technical novelty.
>
> [1] FLUX.1 Kontext: Flow Matching for In-Context Image Generation and Editing in Latent Space
>
> [2] Qwen-Image Technical Report

---

> ### Author Response · Authors · 2025-11-25
> **Response to Reviewer dJ5L (3/n)**
>
> > **W2: Dual-stream architecture vs. Single-stream model. Q2: Explain the motivation for the proposed dual-stream architecture.**
>
> **Answer:**  Our motivation for adopting a dual-stream architecture follows a line of work employing dual-stream designs, including SD3 [1], FLUX [2]. The dual-stream design was first introduced in SD3 as the MM-DiT block, where text tokens and image tokens pass through separate FFNs and separate AdaLN-Zero and modulated with separate (scale, shift, gate params), but still share full self-attention layers and attend to each other through full self-attention. In the SD3 [1] paper, the author names this setting as `MM-DiT(two sets of weights)`.
> The single-stream variant, which uses a unified set of parameters for both modalities, is denoted as `MM-DiT (one set of weight)` or DiT(w/ concatenation of text and image tokens) in SD3 [1].
> In SD3 [1] Section 5.2.3. IMPROVED TEXT-TO-IMAGE BACKBONES, the author directly compare the single stream design vs the dual-stream design and presents results in Figure 4. They show that `MM-DiT(two sets of weights)` significantly outperforms the cross-attention variant and `MM-DiT(one set of weight)` variant. Motivated by this study, we design our model as a two-stream architecture.
>
> To further address the reviewer’s concern, we additionally conducted an ablation study using a single-stream variant. We train the model following the same setting as VOGUE and present the results on video editing are follows:
>
> In Context Insert
>
> | **Model** | **CLIP-I ↑** | **DINO-I ↑** | **CLIP-score ↑** | **Smoothness ↑** | **Dynamic ↑** | **Aesthetic ↑** |
> |----------|-------------|--------------|------------------|------------------|----------------|------------------|
> | **VOGUE(single stream)** |  0.635	| **0.409**	| 0.158	| **0.946**	| 17.602	| 5.703|
> | **VOGUE** | **0.693** | 0.398 | **0.259** | 0.943 | **22.753** | **6.031** |
>
>
> In Context Swap
>
> | **Model** | **CLIP-I ↑** | **DINO-I ↑** | **CLIP-score ↑** | **Smoothness ↑** | **Dynamic ↑** | **Aesthetic ↑** |
> |----------|-------------|--------------|------------------|------------------|----------------|------------------|
> | **VOGUE(single stream)** |  0.633	|  0.269| 	0.197	| **0.973**| 	19.316	| 5.915|
> | **VOGUE** | **0.728** | **0.427** | **0.244** | **0.973** | **19.892** | **6.190** |
>
>
> In Context Delete
>
> | **Model** | **PSNR ↑** | **RefVideo-CLIP ↑** | **CLIP-score ↑** | **Smoothness ↑** | **Dynamic ↑** | **Aesthetic ↑** |
> |----------|-------------|----------------------|------------------|------------------|----------------|------------------|
> | **VOGUE(single stream)** | 13.027	| **0.919** |0.195	|0.966	|17.703	|5.485 |
> | **VOGUE** | **17.980** | 0.888 | **0.214** | **0.971** | **19.502** | **5.498**|
>
>
> Key Observation: Despite using the same data and training settings, we observe a performance drop across most tasks and metrics for the single-stream baseline compared with VOGUE.
>
>
> [1] Scaling Rectified Flow Transformers for High-Resolution Image Synthesis
>
> [2] FLUX.1 Kontext: Flow Matching for In-Context Image Generation and Editing in Latent Space

---

> ### Author Response · Authors · 2025-11-25
> **Response to Reviewer dJ5L (4/n)**
>
> >**W3: The paper provides limited information about the construction of the training dataset. Q2:Could the authors elaborate on the dataset curation pipeline?**
>
> **Answer:**  To address the reviewer’s concern, we have substantially expanded Appendix F, providing a comprehensive description of our data curation pipeline and providing a breakdown of the amount of data required for each task.
>
> >**W5: Missing Citations and comparisons with existing multi-subject video personalization models**
>
> **Answer:** We thank the reviewer for pointing out these relevant works. We have added the appropriate citations and clarified how they differ from VOGUE in the updated submission in both the related work section and the expanded related work section of Appendix C.2.
>
> > **W6: How the model handles out-of-distribution or ambiguous visual prompts.
> Q3: Provide some failure cases to help check the generality of the visual prompt task**
>
> **Answer:** To address the reviewers’ concern, we have included several case studies on the  [anonymous project website](https://anonymous-submission-rebuttal.github.io/vogue/). For ambiguous prompts, for example, simply using arrows to indicate the intended motion of characters. The MLLM component can reasonably interpret the visual prompts, converting them into meaningful semantic instructions to guide the video generation process.
> We also observe limitations of the current model:
> 1. When the number of reference images exceeds the distribution seen during training (maximum of four), the model often fails to incorporate all of them into the generated video. This behaviour is expected given that the model was never trained with larger sets of references. Nonetheless, the generated videos remain high quality, and the model can still robustly incorporate up to four references rather than completely failing on this out-of-distribution task.
> 2. The model struggles with visual prompts requesting multi-shot scene cuts. Since the model is trained only on single-shot videos, multi-shot compositions fall completely outside the training distribution, leading to failure in following such instructions.
> 3. When an image contains many people, each associated with detailed and distinct visual prompts, the model can interpret the instructions but may produce imperfect interactions between individuals. We believe this limitation stems from the capacity of the underlying video generator backbone.
>
> These cases illustrate the boundaries of the current model’s visual prompting understanding capability and provide insights into where future improvements can be made.

---

### Official Review · Reviewer_p7Fb · 2025-10-19

**Soundness:** 3
**Presentation:** 3
**Contribution:** 3
**Rating:** 4
**Confidence:** 4

**Summary:**

The work proposes a method to imbue a pretrained video generator with multimodal input understanding. The work builds on the observation that text embedding can be replaced with embeddings produced by a frozen MLLM, making it possible for the downstream video generator to leverage MLLM capabilities such as thinking and understanding of multimodal inputs. The authors first curate datasets comprising a variety of multimodal tasks such as image editing, image to video, style transfer, object swapping, addition, deletion. Then a pretrained Hunyuan model is adapted to receive QwenVL2.5-7B multimodal inputs in 3 stages, by first training a connector MLP, then fine-tuning the video generator on T2I and T2V, and finally training on the full range of tasks. The resulting model produces convincing results in a variety of tasks and shows ability to generalize outside of the set of training tasks. Quantitative evaluation shows generally better or comparable scores with respect to VACE and commercial models.

**Strengths:**

- The task of creating unified video generators capable of tackling multiple visual tasks from T2V to video editing is of high relevance
- Qualitative results are convincing. The model seems to be able to tackle a range of video generation tasks from object insertion to instruction based editing with convincing quality
- Ablations are insightful and validate that 1) joint training on all tasks reinforces model performance in all tasks, 2) relying on MLLM visual embeddings is insufficient, requiring visual tokens being passed to the video generation backbone directly
- The authors provide complete training and dataset details
- Authors commit to public release of model and code

**Weaknesses:**

- Ablation on MLLM visual embeddings raises doubts of how much the introduction of the MLLM contributes to model performance with respect to the newly collected data. See questions.

Minor:
- Minor amount of typos, especially lack of space before citations (LL254, LL258)
- LL52, LL423-LL427 the claim appears inaccurate. Veo 3 was previously found to support this feature (https://www.reddit.com/r/singularity/comments/1m9b0bq/googles_new_feature_in_veo_3_you_can_now_draw/) (https://www.youtube.com/watch?v=KNGMBRyGcDo). I suggest removing the claim.

**Questions:**

I hope the authors could clarify the following questions
- Table 5 reports that "w/o visual for MLLM" achieves an overall score that is very close to VOGUE. While the exact setting is slightly unclear (are we just removing visual tokens from the MLLM output or is the MLLM not receiving visual tokens at all?) LL440-LL441 suggests that no visual token is processed by the MLLM in this setting. While a small gap is present in the prompt following (PF) metric, this raises the question of whether an MLLM is needed at all. If only text tokens are processed by the MLLM in the performed ablation, wouldn't the original text encoder produce similar performance without the need for an MLLM? This is an important point to clarify as otherwise the role of the MLLM is unclear, and a simpler framework with equal capabilities could be constructed by means of the collected dataset only without any MLLM.
- Table 2 reports a series of Understanding metrics. Do I understand correctly that these metrics are computed using the frozen MLLM model alone without involvement of the video generator? If so, such metrics are less interesting to show. The claim that the model can perform "Understanding" in a unified way with generation and editing in this case would be problematic as the "Understanding" part is completely offloaded to a pretrained and frozen external model, which is not a unified approach. In a truly unified approach, the MM-DiT itself would output text tokens for understanding tasks when necessary.

Minor:
- Could the authors clarify more how the dataset ratios shown in Table 1 were computed? Can the authors offer insights into how optimal ratios should be derived?

---

> ### Author Response · Authors · 2025-11-25
> **Response to Reviewer p7Fb (1/n)**
>
> Dear reviewer `p7Fb`,
>
> We sincerely thank the reviewer for the encouraging and insightful feedback, and for recognizing the high relevance of our unified video generation framework. We are encouraged that the reviewer finds our qualitative results convincing and our ablation studies insightful. We also appreciate the acknowledgment of our commitment to open-sourcing the model and code, which we believe will further benefit the research community.
>
>
> >**W1 & Q1:  "w/o visual for MLLM" achieves an overall score that is very close to VOGUE. whether an MLLM is needed at all. Wouldn't the original text encoder produce similar performance without the need for an MLLM?**
>
> **Answer:** We would like to clarify that the abbreviation “Overall” refers to the "Overall Video Quality" metric, which is defined in Section 3.2.2 Metrics(L269) of our original submission. This metric measures the visual quality of the generated video and is not derived from the Prompt Following or Subject Consistency metrics. We thank the reviewer for pointing this out and acknowledge the ambiguity caused by the abbreviation; we have therefore renamed “Overall” to “VQ” in the updated submission. In Table 5, “w/o visual for MLLM” indicates that the MLLM does not receive visual tokens. We have clarified this in the revised manuscript. The observation that removing visual inputs from the MLLM yields a VQ score close to VOGUE is reasonable, as VQ measures only video quality, and the model can still produce visually good-quality videos even without visual inputs to the MLLM. However, despite a small drop in VQ, we still observe consistent performance declines across the other metrics. we observe clear declines in prompt following, especially for editing tasks that require visual grounding and semantic understanding from the MLLM stream. In particular, the prompt following for task swap drops from 0.91 to 0.75, and the prompt following metric for task delete drops from 0.52 to 0.45. Since the Prompt Following metric directly measures whether an edit is executed correctly, we consider these reductions substantial. Therefore, incorporating visual inputs into the MLLM branch is necessary for accurate prompt following and editing.
>
>
> To further address the reviewer’s concern about whether "an MLLM is needed.", we conducted additional ablation study by training VOGUE without MLLM and using the original text encoders, the same dataset and training settings. This experiment provides a direct answer to whether incorporating an MLLM is necessary.
>
> We will (1) present the quantitative results and (2) provide qualitative comparisons and analysis.
>
> In Context Insert
>
> | **Model** | **CLIP-I ↑** | **DINO-I ↑** | **CLIP-score ↑** | **Smoothness ↑** | **Aesthetic ↑** |
> |----------|-------------|--------------|------------------|----------------|------------------|
> | VACE | 0.513 | 0.105 | 0.103 | 0.947  | 5.693 |
> | UNIC | 0.598 | 0.245 | 0.216 | $\underline{0.961}$ | 5.627 |
> | Kling1.6 | 0.632 | 0.287 | 0.246 | **0.993**  | $\underline{5.798}$|
> | Pika2.2 | $\underline{0.692}$| **0.399** | $\underline{0.253}$| 0.951 | 5.591 |
> | **VOGUE w/o MLLM** |  0.679	| 0.325	| 0.232 |	0.959	|  5.981 |
> | **VOGUE** | **0.693** | $\underline{0.398}$ | **0.259** | 0.943 | **6.031** |
>
>
> In Context Swap
>
> | **Model** | **CLIP-I ↑** | **DINO-I ↑** | **CLIP-score ↑** | **Smoothness ↑**  | **Aesthetic ↑** |
> |----------|-------------|--------------|------------------|----------------|------------------|
> | VACE | 0.703 | 0.391 | 0.218 | 0.960 | 5.961 |
> | UNIC | $\underline{0.725}$ | $\underline{0.429}$ | $\underline{0.242}$ | 0.971 | $\underline{6.056}$ |
> | Kling1.6 | 0.707 | **0.437** | 0.211 | **0.995**  | 6.042 |
> | Pika2.2 | 0.704 | 0.406 | 0.211 | 0.967 | 5.097 |
> | AnyV2V | 0.605 | 0.229 | 0.218 | 0.917 | 4.842 |
> | **VOGUE w/o MLLM** |  0.645	| 0.318	| 0.227	| 0.968	| 6.043|
> | **VOGUE** | **0.728** | 0.427 | **0.244** | $\underline{0.973}$ | **6.190** |
>
>
> In Context Delete
>
> | **Model** | **PSNR ↑** | **RefVideo-CLIP ↑** | **CLIP-score ↑** | **Smoothness ↑**  | **Aesthetic ↑** |
> |----------|-------------|----------------------|------------------|----------------|------------------|
> | VACE | $\underline{20.601}$| 0.874 | 0.206 | 0.968  | **5.637** |
> | UNIC | 19.171 | 0.817 | **0.217** | 0.970 | 5.493 |
> | Kling1.6 | 15.476 | $\underline{0.888}$| 0.208 | **0.998**  | 4.965 |
> | AnyV2V | 19.504 | 0.869 | 0.205 | 0.964  | 5.325 |
> | VideoPainter | **22.987** | **0.920** | 0.212 | 0.957  | 5.403 |
> | **VOGUE w/o MLLM** | 11.202 |  0.816  | 0.196 | $\underline{0.971}$ | 5.385 |
> | **VOGUE** | 17.980 | $\underline{0.888}$ | $\underline{0.214}$ | $\underline{0.971}$ | $\underline{5.498}$ |

---

> ### Author Response · Authors · 2025-11-25
> **Response to Reviewer p7Fb (2/n)**
>
> Single Reference Generation
>
> | **Model** | **SC ↑** | **PF ↑** | **VQ ↑** | **Smoothness ↑** | **Aesthetic ↑** |
> |----------|----------|----------|----------------|------------------|------------------|
> | VACE | 0.31 | 0.65 | 0.42 | 0.922  | 5.426 |
> | Kling1.6 | 0.68 | **0.95** | $\underline{0.88}$ | $\underline{0.938}$ | **5.896** |
> | Pika2.2 | 0.45 | 0.43 | 0.15 | 0.928 | 5.125 |
> | **VOGUE w/o MLLM** | $\underline{0.81}$ | 0.71 | 0.86 |  0.925 | 5.725 |
> | **VOGUE** | **0.88** | $\underline{0.93}$ | **0.95** | **0.943** | $\underline{5.740}$ |
>
> Multi Reference Generation
>
> | **Model** | **SC ↑** | **PF ↑** | **VQ ↑** | **Smoothness ↑** | **Aesthetic ↑** |
> |----------|----------|----------|----------------|------------------|------------------|
> | VACE | 0.48 | 0.53 | 0.48 | 0.862 | 5.941 |
> | Kling1.6 | 0.73 | 0.45 | **0.95** | 0.916  | $\underline{6.034}$ |
> | Pika2.2 | 0.71 | 0.48 | 0.43 | 0.898 | 5.176 |
> | **VOGUE w/o MLLM** | $\underline{0.76}$  | $\underline{0.69}$ | 0.78 |  $\underline{0.932}$ | 6.006 |
> | **VOGUE** | **0.81** | **0.75** | $\underline{0.85}$ | **0.942** | **6.128** |
>
> Key Observation: Despite using the same video generation backbone and the same data, we observe a performance drop across most of the tasks and metrics compared with VOGUE.
>
> **Analysis:**
> **We also provide qualitative comparisons on the [anonymous project website](https://anonymous-submission-rebuttal.github.io/vogue/)**
>
> Our analysis shows that the MLLM is particularly important for tasks requiring strong visual grounding.
> We provide qualitative examples when the model handles challenging multi-reference inputs, such as when the reference image is not a close-up shot of a single object and instead may contain multiple objects with different spatial arrangements, and the model is required to attend only to the specific objects indicated in the prompt. This setting demands a strong visual understanding and grounding ability to correctly identify which instance the user refers to. The VOGUE w/o MLLM baseline often fail in such cases.
> Additionally, in editing tasks that require fine-grained grounding—such as deleting a small object at the border of the frame (e.g., a clock on the wall), or swapping an object at the edge of the video (e.g., a paper bag on the floor), or tasks requiring prior visual knowledge (e.g., replacing an object with Pikachu). The VOGUE w/o MLLM baseline consistently fails to follow these instructions, whereas VOGUE succeeds.
>
> Last but not least, the incorporation of an MLLM also enables the model to achieve visual prompt understanding ability, as discussed in Section 3.3.2 THINKING MODE, which cannot be accomplished using only the original text encoder.

---

> ### Author Response · Authors · 2025-11-25
> **Response to Reviewer p7Fb (3/n)**
>
> >**Q2 Understanding metrics are reported using the frozen MLLM model alone**
>
> **Answer:** We thank the reviewer for raising this question. We have added clarification that "We report understanding task results for VOGUE
> using the MLLM component — Qwen-2.5VL-7B results" in the updated submission.
>
> Our work follows the line of literature on unified understanding and generation in the text and image domains, including MetaQueries [1], OmniGen2 [2], Uniworld [3], BLIP3-o [4], UniCTokens [5], and OpenUni [6]. This line of post-training research builds unified models by bridging a pretrained (frozen) large vision–language model with a generative model, instead of pretraining a new unified model, which is computationally expensive in terms of both data and GPU requirements. To be consistent with these works, which also report understanding performance using a frozen VLM, we report Table 2 using a frozen vision language model following this established paradigm. **Models in this category are viewed as post-trained assembled multimodal generative models** [2], rather than models trained end-to-end from scratch. To further address the reviewer’s concern, we have updated the submission and now explicitly highlight at the end of the Introduction section that "VOGUE’s text generation capability originates from a frozen MLLM, and thus VOGUE should be regarded as a post-trained unified multimodal model". **Nevertheless, VOGUE possesses both the visual understanding and visual generation capabilities expected of a unified model**. It can perform a reasoning process through the MLLM stream to interpret user intent from multimodal instructions and subsequently guide the video generator stream accordingly, as demonstrated by the visual prompt understanding results presented in the paper and on the project website. We believe future work can build upon VOGUE to support additional multimodal generation tasks.
>
> **Compared with pretraining systems such as JanusFlow [7], Transfusion [8], Emu3 [9], and BAGEL [10], which require hundreds to thousands of GPUs to train, post-trained models offer similar capabilities while being far more lightweight.**
> Pretraining a unified text–image model from scratch would be prohibitively expensive, and the cost would be even higher for a model with full video generation capability. In contrast, our method provides an efficient and practical solution: **training VOGUE requires only 32 GPUs, a fact noted by Reviewer 2yjX**:
> "The results are really good, especially given the little amount of GPUs used to fine-tune the model. Also, the data size is affordable to collect for non-bigtech companies."
> Furthermore, while prior progress has focused on text–image unified modeling, our work is among the first to extend unified modelling to the video domain, covering a broad set of tasks in a single framework.
> We believe **VOGUE provides a valuable open-source baseline** for future unified video model research. Future work may compare fully end-to-end, resource-heavy unified video models against this baseline.
>
>
> Although recent large-scale pretraining models have advanced multimodal learning, **studies on why unified models are necessary remain limited**. Our work directly demonstrates the benefits of unified modeling through two emergent synergies:
>
> (a)First, VOGUE enables two forms of generalization. Figure 5 and demos on the website show VOGUE
> generalize to task composition, such as combining editing with style transfer, by integrating multiple capabilities within a single instruction.
> Second, even without explicit training on free-form video editing, VOGUE transfers its editing capability from large-scale image editing data to this setting and generalizes to unseen instructions, such as changing the environment or changing materials within a video.
>
> (b)Second, our ablation in Table 5 further shows that multi-task training improves performance on every task compared to single-task experts. Specifically, we compare VOGUE with single-task baselines. Single-task baselines share the same architecture as VOGUE but require an independent model for each task and have access only to task-specific data. VOGUE consistently outperforms the baseline on all tasks, which highlights the importance of joint task learning.

---

> ### Author Response · Authors · 2025-11-25
> **Response to Reviewer p7Fb (4/n)**
>
> Lastly, we would like to **highlight that effectively aligning a vision–language model with a video diffusion generator is nontrivial.** Although many works align VLMs with image generators, we find it not easy to transfer those methods directly to the video domain. In our updated submission, we add Appendix G (Model Design) to describe several alternative alignment strategies we attempted:
>
> (1)using an MLP to align a pretrained MLLM with a cross-attention DiT, training only the MLP;
>
> (2)using learnable queries to align with a cross-attention DiT, training only the queries;
>
> (3)using learnable queries with a cross-attention DiT, training both the queries and the DiT;
>
> (4)our final VOGUE architecture: using an MLP to align a pretrained MLLM with a self-attention MM-DiT, where text–video interaction occurs through self-attention rather than cross-attention.
>
> Our findings show that the proposed VOGUE design is the most effective among these options: under the same training steps, VOGUE converges the fastest and achieves the best efficiency with minimal trainable parameters. We believe these findings provide practical and actionable insights for the open-source community on how to efficiently build unified video models during post-training.
>
>
> [1] MetaQueries: Transfer between Modalities with MetaQueries
>
> [2] OmniGen2: Exploration to Advanced Multimodal Generation
>
> [3] Uniworld: High-resolution semantic encoders for unified visual understanding and generation
>
> [4] Blip3-o: A family of fully open unified multimodal models-architecture, training and dataset
>
> [5] UniCTokens: Boosting Personalized Understanding and Generation via Unified Concept Tokens
>
> [6] OpenUni: A Simple Baseline for Unified Multimodal Understanding and Generation
>
> [7] JanusFlow: Harmonizing Autoregression and Rectified Flow for Unified Multimodal Understanding and Generation
>
> [8] Transfusion: Predict the Next Token and Diffuse Images with One Multi-Modal Model
>
> [9] Emu3: Next-Token Prediction is All You Need
>
> [10] BAGEL: Emerging Properties in Unified Multimodal Pretraining

---

> ### Author Response · Authors · 2025-11-25
> **Response to Reviewer p7Fb (5/n)**
>
> >Minor W1 Typos, especially lack of space before citations (LL254, LL258)
>
> **Answer:** We thank the reviewer for the detailed guidance. We have corrected these typos in the revised manuscript.
>
> >Minor W2 Veo3 was previously found to support this feature. Removing the claim (LL52, LL423-LL427).
>
> **Answer:** We thank the reviewer for the suggestion. We have removed the original claim, rephrased the corresponding sentences, and now cite Veo3 to reflect its support for the second type of visual prompt, where annotations are drawn directly on the input image. Our study provides the open source community with a potential solution for achieving similar capabilities to those of the closed-source model.

---

### Official Review · Reviewer_2yjX · 2025-10-30

**Soundness:** 3
**Presentation:** 3
**Contribution:** 3
**Rating:** 8
**Confidence:** 4

**Summary:**

The paper presents a system for joint image/video understanding, generation and editing. It consists on two streams: understanding and generation. The understanding stream is the frozen Qwen-VL2.5-7B. The generation stream is initialized from HunyuanVideo-T2V-13B. The training is split into 3 stages: 1) training the connector between MLLM and MMDiT; 2) fine-tuning MMDiT for base tasks (T2I + T2V); 3) multi-task fine-tuning of MMDiT. The results look very good visually, and the model was even shown to zero-shot generalize to novel tasks (e.g. free-form video editing). The authors also perform a thorough quantitative evaluation demonstrating the superiority of the method over existing baselines.

**Strengths:**

- It is a quite elegant architecture which i think among the early works for this paradigm shift in generative modeling of concatenating everything into a single sequence and modeling. While it does not serve as a good reference for ablations, but it's a good proof of concept to convince the community that this paradigm works.
- The results are really good, especially given the little amount of GPUs used to fine-tune the model. Also, the data size is affordable to collect for non-bigtech companies (e.g. startups)
- The paper reads well and the illustrations are good. The submission also includes many qualitatives which are very easy to view on the website.

**Weaknesses:**

- From my perspective, the main weakness is the lack of rich ablations, that could make the submission to be a good reference for follow up works. For example, is it necessary to do all the 3 stages sequentially or we can train jointly? Is the diffusion schedule the same for all the modalities? How much improvement can we get by making it different? Would it help to do some dropout on some input modalities in a task? Is it possible to replace full attention with cheaper attention variants in-between the modalities? Also some profiling results would be interesting to see, e.g. how much compute is spent on each modality in each component (is MLLM heavy?) And so on.
- Some important details are missing from the submission, mainly related to data curation. It would be fine to omit them for a technical report, but it is not good practice to omit them for an academic submission. For example, how exactly was stylized video transformed into a real one (appendix F.2)? How was the video inpainting model training (appendix F.1)? Was the dataset post-curated with human annotators? If so, what were the instructions for them?

Small writing comments:
- typo: "generate an video" => "generate a video" (in figures)
- typo: "source open source" on line 907
- I would suggest re-coloring Figure 3 to display the frozen MLLM as blue, and trainable DiT as red, as common in the prior literature.

**Questions:**

Could you please include more dataset details (as specified in the previous section)?

---

> ### Author Response · Authors · 2025-11-25
> **Response to Reviewer 2yjX (1/n)**
>
> Dear reviewer `2yjX`,
>
> We sincerely thank the reviewer for the thoughtful and constructive feedback, as well as for the positive assessment of our model design, presentation, and the strong visual results. Below, we address each concern in detail and clarify the changes we made in the revised version (marked in violet).
>
>
> >**W2 & Q1: include more dataset curation details**
>
> **Answer:** To address the reviewer’s concern, we have substantially expanded Appendix F, providing a comprehensive description of our data curation pipeline and addressing all questions in detail. For clarity, we also provide direct answers to each of the reviewer’s points below.
>
> >How exactly was stylized video transformed into a real one (appendix F.2)?
>
> The input to the ControlNet is a gray tile signal. Specifically, we downsample the video spatially by a factor of 8 and then upsample it by the same factor to remove high-frequency details, producing a low-fidelity tile image. We further discard the color information by converting this tile image into grayscale. This results in a structural guidance signal that preserves spatial layout while suppressing style and texture.
>
> Similar to StyleMaster, the ControlNet is built on a 1B-parameter DiT architecture similar to Wan2.1, which combines cross-attention for text conditioning with self-attention over visual tokens. We construct the ControlNet by copying an interleaved half of the Transformer blocks from the original DiT. While the original DiT processes noisy video tokens alongside text tokens, the ControlNet blocks operate on the gray tile signal together with the text tokens. The output of each ControlNet block is injected back into the DiT through additive residual connections.
>
> We train the stylized-to-real ControlNet using 10K video pairs in which both the input and target videos are real. During training, the model therefore learns a real-to-real reconstruction task. Since the control signal (the gray tile) preserves only coarse spatial structure while discarding color, details, and style, the model learns to generate realistic content guided only by spatial layout. At inference time, the model can effectively perform stylized-to-real mapping because the stylized input video is also converted into a gray-tile signal, which contains only spatial layout information and thus matches the training distribution.
>
> >How was the video inpainting model training (appendix F.1)?
>
> The inpainter is built on a 1B-parameter model with an architecture similar to Wan2.1, which employs cross-attention modules for text conditioning and self-attention for visual tokens. We copy an interleaved half of the Transformer blocks from the original DiT to form the control net. While the original DiT processes noisy video tokens together with text tokens, the newly added control blocks operate on the masked video, the corresponding masks, and the text tokens. The output of each control block is injected back into the DiT as an additive control signal.
>
> To train the video inpainter, we use the open source dataset VIVID-10M, which provides source video and image and object mask for inpainter training.
>
>
> >Was the dataset post-curated with human annotators? If so, what were the instructions for them?
>
> Yes. After constructing the dataset, we conduct a human filtering stage to ensure the final quality of videos. Annotators are provided with both the source video and the edited video and evaluate each sample solely based on three criteria: video quality, instruction following, and consistency with the source video(degree of overedit)
>
> For object removal and addition tasks, a sample is accepted only if the edit satisfies all three dimensions: (1) high video quality, meaning the edited region is clear and artifact-free; (2) correct execution of the instruction, such as fully removing or appropriately adding the target object; and (3) consistency with the original video, ensuring natural backgrounds and no over-editing beyond the target region. Any sample exhibiting artifacts, partial edits, or temporal flicker is rejected.
>
> For object swap tasks, annotators apply the same three metrics. A sample is accepted only if (1) the edited content is visually stable and free of distortions, (2) the swap operation correctly follows the instruction, and (3) the resulting video remains consistent with the original motion, lighting, and scene dynamics. Samples containing structural distortions, unnatural textures, or temporal inconsistency are rejected. Identity verification is unnecessary, as the source video already defines the intended target appearance.

---

> ### Author Response · Authors · 2025-11-25
> **Response to Reviewer 2yjX (2/n)**
>
> >**Small writing comments.**
>
> **Answer:** We thank the reviewer for the helpful suggestion. We have corrected the mentioned typos in the updated version.
>
> >**W1. Including more ablations.**
>
> **Answer:** We thank the reviewer for the helpful suggestion.
> We have added several ablation studies in Appendix G, including (1) ablations on the most effective methods to align an MLLM with the video diffusion model during Stage 1 training, (2) ablations evaluating whether using an MLLM provides benefits over simple text encoders. We are continuing to work on additional ablation studies and will keep the submission updated accordingly.
>
> In particular, we would like to emphasize our ablation study on the most effective strategies for aligning an MLLM with a video diffusion model during Stage 1 training. We found aligning a vision–language model with a video diffusion generator is nontrivial. Although many works align VLMs with image generators, we find it not easy to transfer those methods directly to the video domain. In our updated submission, we add Appendix G.1 (Model Design) to describe several alternative alignment strategies we attempted:
>
> (1)using an MLP to align a pretrained MLLM with a cross-attention DiT, training only the MLP;
>
> (2)using learnable queries to align with a cross-attention DiT, training only the queries;
>
> (3)using learnable queries with a cross-attention DiT, training both the queries and the DiT;
>
> (4)our final VOGUE architecture: using an MLP to align a pretrained MLLM with a self-attention MM-DiT, where text–video interaction occurs through self-attention rather than cross-attention.
>
> Our findings show that the proposed VOGUE design is the most effective among these options: under the same training steps, VOGUE  achieves the best efficiency with minimal trainable parameters. We believe these findings provide practical and actionable insights for future research.

---

> > ### Comment · Reviewer_2yjX · 2025-11-27
> >
> > I am thankful to the authors for answering my questions, and for providing additional details and ablations. I chose to keep my original score, and not increase it to 10 (oral publication), because there are no particularly novel insights in the paper, it is more of a solid technical system, serving as a good reference to the community.

---

### Official Review · Reviewer_zojR · 2025-11-01

**Soundness:** 3
**Presentation:** 3
**Contribution:** 3
**Rating:** 6
**Confidence:** 3

**Summary:**

The paper is cleanly structured and easy to read. Qualitative results are abundant and illustrative, and the in-context editing/generation comparisons are clearly presented. At a high level, the work’s strengths are system-level integration and breadth of tasks. The level of novelty is average but acceptable.

**Strengths:**

1. A pragmatic dual-stream unification (frozen MLLM for understanding + MMDiT for generation) that feeds visual inputs to both streams, with ablations showing why both sides need visuals for identity preservation and semantics.
2. Strong mask-free in-context editing/generation results despite baselines using masks; the qualitative figures and automatic metrics make the claim legible and practically relevant.
3. Transparent training recipe and task coverage, including staged training, freezing choices, connector design, and explicit mixing ratios. It is useful for reproduction and future baselines.

**Weaknesses:**

1. Thinking Mode is not well and fully discussed(How much does it benefit?).
2. What is the benefit of making it a single model rather than using a workflow or agent? Qualitative analysis?

**Questions:**

1. The workflow diagram (Fig. 3) isn’t very clear. Is it correct to interpret that the user feeds an interleaved text–image instruction to the MLLM, and then the MLLM’s output is concatenated with the VAE features of the conditioning images as the input? Also, why is the noise term missing in Fig. 3? It feels like this should be made consistent with Fig. 2.
2. Maybe compare to more unified understanding & generation model

---

> ### Author Response · Authors · 2025-11-25
> **Response to Reviewer zojR (1/n)**
>
> Dear reviewer `zojR`,
>
> We sincerely thank the reviewer for the thoughtful and constructive feedback, as well as for the positive assessment of our model design, presentation, and the strong results achieved by our unified multimodal model. Below, we address each concern in detail and clarify the changes we made in the revised version (marked in violet).
>
> >**W1 Thinking Mode is not well and fully discussed(How much does it benefit?)**
>
> **Answer:** To address the reviewer’s concern, we conducted additional experiments comparing VOGUE(Without Thinking Mode) vs. VOGUE on visual prompt understanding.
> VOGUE without Thinking: Only the system prompt and the single canvas image containing the visual prompt are fed into the MLLM. The hidden states of this image are then provided to the MMDiT. The MMDiT also receives the reference images (which contain ID information) encoded by the VAE.
>
> VOGUE with Thinking: The single canvas image containing the visual prompt is first fed into the MLLM, which is instructed to understand the visual prompt and explicitly generate a coherent dense caption to guide video generation.
> Then, the hidden states of the canvas image and MLLM's output and reference images (encoded by VAE) are as inputs for the MMDiT.
>
> Following the evaluation protocol described in Section 3.2.2 Metrics, we construct a test set containing diverse visual prompts. Each sample is rated by at least three annotators on (i) subject consistency (SC), (ii) prompt following (PF), and (iii) overall video quality (VQ).
>
> Our results are as follows, and we also include qualitative comparisons on the [anonymous project website](https://anonymous-submission-rebuttal.github.io/vogue/)
>
> | Method  |PF↑| SC↑| VQ↑ |
> |-|-|-|- |
> | VOGUE    |0.71| 0.81| 0.83 |
> | VOGUE without thinking  |0.21| 0.57| 0.58 |
>
> Analysis: Our results show that thinking mode significantly enhances the generation quality especially in the prompt following. Although the model without thinking can still produce meaningful video, it often fails to strictly follow the prompt when the visual prompt’s textual symbols encode temporal order or event sequencing. Thinking Mode generates a dense and coherent caption that makes these temporal cues explicit, providing a much reliable control signal for the video generation component.
>
>
> > **W2: What is the benefit of making it a single model rather than using a workflow or agent? Qualitative analysis?**
>
> **Answer:** To address the reviewer’s concern, we provide a detailed analysis of why developing a single unified model offers advantages over workflow- or agent-based pipelines.
>
> 1. Benefits of a Single Unified Model
>
> As a single model, multiple tasks can be trained jointly.
> We observed promising synergy effects:
> (a)First, VOGUE enables two forms of generalization. Figure 5 and demos on the website show VOGUE
> generalize to task composition, such as combining editing with style transfer, by integrating multiple capabilities within a single instruction.
> Second, even without explicit training on free-form video editing, VOGUE transfers its editing capability from large-scale image editing data to this setting and generalizes to unseen instructions, such as changing the environment or changing materials within a video.
> (b)Second, our ablation in Table 5 further shows that multi-task training improves performance on every task compared to single-task experts. Specifically, we compare VOGUE with single-task baselines. Single-task baselines share the same architecture as VOGUE but require an independent model for each task and have access only to task-specific data. VOGUE consistently outperforms the baseline on all tasks, which highlights the importance of joint task learning.
>
> In contrast, workflow or agentic systems typically rely on an LLM to route requests to multiple task-specific expert models. These systems cannot benefit from shared representations or cross-task transfer. VOGUE instead demonstrates the potential of a single model that effectively handles diverse vision tasks, providing insight for future research.
>
>
> 2. Qualitative comparison with a commercial agentic system.
>
> We further provide a comparison on visual prompt understanding tasks with the commercial agentic product. Higgsfield provide a "Draw to Video" feature which allows users to upload images into a canvas, make annotations and generate a video, which shares similar settings to our visual prompt understanding tasks.
> We provided a direct qualitative comparison of VOGUE vs. Higgsfield output on our website.
>
> Observations: (1)Higgsfield’s output typically reproduces the input canvas almost verbatim during the first 1–2 seconds. (2) It sometimes misses important visual references (e.g., omitting the man riding the motorcycle). (3) Our method shows comparable or better visual prompt understanding and can generate outputs with more coherent storytelling.

---

> ### Author Response · Authors · 2025-11-25
> **Response to Reviewer zojR (2/n)**
>
> >Q1 The noise term is missing in Fig. 3
>
> **Answer:** We thank the reviewer for the helpful suggestion. We have revised Fig. 3 to improve clarity and added the noise video term for consistency with Fig. 2. The interpretation is correct — the user provides an interleaved text–image instruction to the MLLM, whose output is concatenated with the VAE features of the conditioning images as input.
>
>
> >Q2 Maybe compare to more unified understanding & generation model
>
> **Answer:** We thank the reviewer for the helpful suggestion. In the updated submission, we have included additional unified understanding and generation baselines in Table 2. It is worth noting that existing works are primarily unified image understanding and generation models. To the best of our knowledge, VOGUE is among the earliest unified video models that support a wide range of video tasks

---

### Author Response · Authors · 2025-12-04
**Summary of Rebuttal Updates**

Dear Area Chair,

Given the recent administrative reset of review states, we provide this summary to assist your assessment.

### **1. All Reviewer Concerns Addressed with Highlighted Revisions**

We addressed all concerns in our responses, and the corresponding revisions are highlighted in **violet** in the revised submission.

In particular:

1. Dataset Construction Pipeline

We have enriched the dataset construction pipeline with additional details in Appendix F.

2. Additional Experiments Requested by Reviewers

We have included additional experiments in each response and in Appendix G, and we also provide qualitative results on the anonymous website.

### **2. We summarize the strong positive consensus expressed by the reviewers below.**

- **Position of VOGUE among the earliest works on unified video models, covering a broad set of tasks:**

> **RdJ5L:**  *"As far as I understand, VOGUE is the first video generation model that achieves this level of task unification, which can substantially enhance the community interest."*
> **Rp7Fb:**  *"The task of creating unified video generators capable of tackling multiple visual tasks from T2V to video editing is of high relevance."*

- **We also highlight the novelty of the proposed architecture:**

> **R2yjX:**  *"It is a quite elegant architecture which I think is among the early works for this paradigm shift in generative modeling."*
> **RzojR:**   *"A pragmatic dual-stream unification."*

For **RdJ5L’s concern**,
We would like to clarify that the Qwen-Image preprint was released on `August 4, 2025`, while our work was submitted on `September 19, 2025`. The two works therefore represent **concurrently developed efforts** (within two months). While Qwen-Image explores a related design idea in the **image editing** setting, our model instead targets the **video domain** and a **multitask unified** setting.

With this context, our work still provides distinct insights and contributions by:

1. Justifying the necessity of **MLLM embeddings**
2. Studying how to **efficiently align MLLMs with the video generator**
3. Studying how to **develop a unified model and demonstrate emergent generalization capabilities**

Reviewers find these studies insightful:

> **Rp7Fb:** *"Ablations are insightful."*
> **RdJ5L:** *"The authors also provide thorough ablation studies, which are insightful."*

- **Reviewers are convinced by our qualitative results:**

> **RzojR:** *"The qualitative figures and automatic metrics make the claim legible and practically relevant."*
> **R2yjX:** *"The results are really good."*
> **Rp7Fb:** *"Qualitative results are convincing."*

- **Reviewers also recognize the value of our open-source model**, which is useful for reproduction and future baselines:

> **RzojR:** *"Transparent training recipe. It is useful for reproduction and future baselines."*
> **Rp7Fb:** *"The authors provide complete training and dataset details."*
> **RdJ5L:** *"The authors promise that they will release the model checkpoint and code, which can significantly increase the reproducibility and also benefit further research in unified video generation and editing."*

Together, these open-source assets provide the **essential infrastructure for the open-source community to build unified video models and help bridge the gap with closed-source systems**.

---

### Meta-Review · Area_Chair_kzp2 · 2026-01-07

**Summary:**

The paper proposes VOGUE, a unified model for the understanding, generation, and editing of videos. It uses an MLLM to process multimodal inputs, whose output is used as a condition to the diffusion transformer. The model is trained in multiple stages. The model is validated on a diverse set of tasks and shows impressive results.

There are also several concerns pointed out by the reviewers, which have been addressed during the rebuttal phase. Therefore, I would recommend acceptance of this work. I encourage the authors to incorporate reviewers' suggestions in their next version.

**Reviewer Concerns:**

Most of the major concerns are addressed by the rebuttal.

**Reviewer Scores:**

6, 8, 4, 6 -> 6, 8, 4, 6

---

### Decision · Program_Chairs · 2026-01-26

Accept (Poster)